# Syntactic and Semantic Control of Large Language Models via Sequential Monte Carlo

**João Loula**[*1] **Benjamin LeBrun**[*5] **Li Du**[*6] **Ben Lipkin**[1] **Clemente Pasti**[2] **Gabriel Grand**[1]
**Tianyu Liu**[2] **Yahya Emara**[2] **Marjorie Freedman**[8] **Jason Eisner**[6] **Ryan Cotterell**[2]
**Vikash Mansinghka**[‡1] **Alexander K. Lew**[‡1,7] **Tim Vieira**[‡2] **Timothy J. O'Donnell**[‡3,4,5]
[1]MIT  [2]ETH Zürich  [3]McGill  [4]Canada CIFAR AI Chair  [5]Mila  [6]Johns Hopkins  [7]Yale  [8]ISI
genlm@mit.edu

## Abstract

A wide range of LM applications require generating text that conforms to syntactic or semantic constraints. Imposing such constraints can be naturally framed as *probabilistic conditioning*, but exact generation from the resulting distribution—which can differ substantially from the LM's base distribution—is generally intractable. In this work, we develop an architecture for controlled LM generation based on sequential Monte Carlo (SMC). Our SMC framework allows us to flexibly incorporate domain- and problem-specific constraints at inference time, and efficiently reallocate computational resources in light of new information during the course of generation. By comparing to a number of alternatives and ablations on four challenging domains—Python code generation for data science, text-to-SQL, goal inference, and molecule synthesis—we demonstrate that, with little overhead, our approach allows small open-source language models to outperform models over $8\times$ larger, as well as closed-source, fine-tuned ones. In support of the probabilistic perspective, we show that these performance improvements are driven by better approximation to the posterior distribution. Our system builds on the framework of Lew et al. (2023) and integrates with its *language model probabilistic programming language*, giving users a simple, programmable way to apply SMC to a broad variety of controlled generation problems.

 https://github.com/probcomp/genlm-control

## 1 Introduction

The goal of *controlled generation* from language models (LMs) is to produce text guided by a set of syntactic or semantic constraints. One prominent example is semantic parsing, or code generation, which involves producing text in a programming (or other formal) language, typically from a natural language prompt. We may wish to use diverse signals to guide code generation, for example:

- Checking (partial) code statically (type-checking, linting, partial evaluation);
- Running (partial) code on a test case and checking if it raises an error or returns the wrong answer;
- Simulating environments (e.g., in robotics or chemistry) and assigning a score to the resulting state;
- Rolling out possible completions of partial code and computing their max, min, or average score;
- Asking another language model to critique the code generated so far.

Such signals vary along several important dimensions: some are cheap to compute (linting), others are more costly (simulations); some can be evaluated incrementally with each sampled token (language model critique), others provide sparser guidance (running code); some enforce binary hard constraints (type-checking), others yield soft continuous scores (scoring).

One way to represent such signals uniformly is as *potential functions* $\phi(\boldsymbol{x})$ assigning non-negative scores to sequences of tokens $\boldsymbol{x}$. Given a set $\Phi$ of such potentials, we will write $\Phi(\boldsymbol{x}) = \prod_{\phi \in \Phi} \phi(\boldsymbol{x})$. We frame the problem of controlled generation probabilistically: We wish to sample from the *global*

---

[*]co-first authorship, [‡]co-senior authorship.

*product of experts* distribution on complete sequences $\boldsymbol{x}$:

$$g(\boldsymbol{x}) = \frac{1}{Z}\, p(\boldsymbol{x})\Phi(\boldsymbol{x}) \tag{1}$$

where $p$ is a distribution over complete token sequences defined by an autoregressive LM, and $Z$ is a normalizing constant. The distribution $g$ can be interpreted as a posterior with $p$ as the prior and $\Phi$ as the likelihood function. Importantly, even when both sampling from $p$ and evaluating $\Phi(\boldsymbol{x})$ are efficient, sampling exactly from $g$ is generally intractable (Rosenfeld et al., 2001).

Two popular sampling-based approaches that avoid the intractability of $g$ are *locally constrained decoding* and *sample reranking* (see Appendix F for a detailed review of related work, including other approaches such as MCMC). Locally constrained decoding uses per-token logit biasing or masking to incorporate signals at each step, for example to ensure that the complete sequence will fall in a specified regular or context-free language (e.g., Shin et al., 2021; Scholak et al., 2021; Poesia et al., 2022; Willard & Louf, 2023; Moskal et al., 2024; Ugare et al., 2024). Sample-rerank-based approaches first generate complete sequences and then rerank or reweight these based on the specified set of signals. Examples of this approach include best-of-$n$ reweighting with a reward model (Nakano et al., 2021; Krishna et al., 2022; Zhou et al., 2023; Gui et al., 2024; Mudgal et al., 2024; Ichihara et al., 2025) or filtering samples with a verifier (Olausson et al., 2023; Chen et al., 2024; Lightman et al., 2024; Xin et al., 2024). Each approach suffers from significant weaknesses. Locally constrained decoding requires the guiding signals $\Phi$ to be cheap enough to evaluate very frequently (for instance on the full token vocabulary at every step of generation). Moreover, it often introduces greedy approximations that badly distort the distribution (relative to $g$, Lew et al., 2023; Park et al., 2025). Sample-rerank does not impose constraints $\Phi$ until full sequences have been sampled and, thus, cannot make use of information available incrementally during generation; this can significantly increase the number of samples needed to find high probability, constraint-satisfying sequences.

*Sequential Monte Carlo* (SMC) has been proposed as an effective approach to approximate inference for such intractable distributions in other difficult language modeling problems, such as infilling, prompt engineering, and prompt intersection as well as for more traditional tasks in natural language processing (Börschinger & Johnson, 2011; Dubbin & Blunsom, 2012; Buys & Blunsom, 2015; Lin & Eisner, 2018; Lew et al., 2023; Zhao et al., 2024). In this paper, we use SMC to tackle a number of challenging semantic parsing problems, guiding generation with incremental static and dynamic analyses. When these signals are efficient enough to be used incrementally, our approach incorporates them into *proposal distributions*, gaining the benefits of locally constrained decoding; more costly potentials—as often used in sample-rerank approaches—are incorporated as *twist functions* that reweight partial sequences to favor promising paths (Naesseth et al., 2019). Our approach emphasizes *programmable* potentials and proposals that can easily be specialized for specific tasks or problems (e.g., by integrating libraries for molecular structure or robotic planning, see §3.1). We contrast this with the use of *learned* proposals or twist functions (Lawson et al., 2022; Zhao et al., 2024), which requires costly, problem-specific fine-tuning.

Our paper makes the following contributions:

- *SMC for constrained semantic parsing.* We develop an architecture specializing SMC for code generation under diverse syntactic and semantic constraints (§2). Unlike many previous frameworks for constrained decoding, our algorithm can integrate constraints that cannot be incrementally evaluated over the entire token vocabulary, as well as constraints that can only be evaluated at irregular intervals during generation. The framework emphasizes *programmable inference* (Mansinghka et al., 2018), allowing users to deploy proposals and potentials that exploit the structure of application domains. Our system also fully integrates with the language model probabilistic programming framework of Lew et al. (2023).
- *Empirical evaluation of performance in diverse domains.* We apply our approach—and six alternatives—to four challenging problem domains: Python code generation for data science, text-to-SQL, goal inference, and molecule synthesis (§3.1). We find that, with little overhead, our approach significantly improves performance across domains, allowing small open-source language models to outperform models over $8\times$ larger, as well as closed-source fine-tuned models. We additionally find that, with $5$–$10\times$ fewer particles, SMC outperforms approaches that only incorporate constraints at the end of generation (§3.2).
- *Empirical evaluation of algorithm components.* We run ablation experiments and find that improved performance can be attributed to three algorithmic components: *weight correction*, which

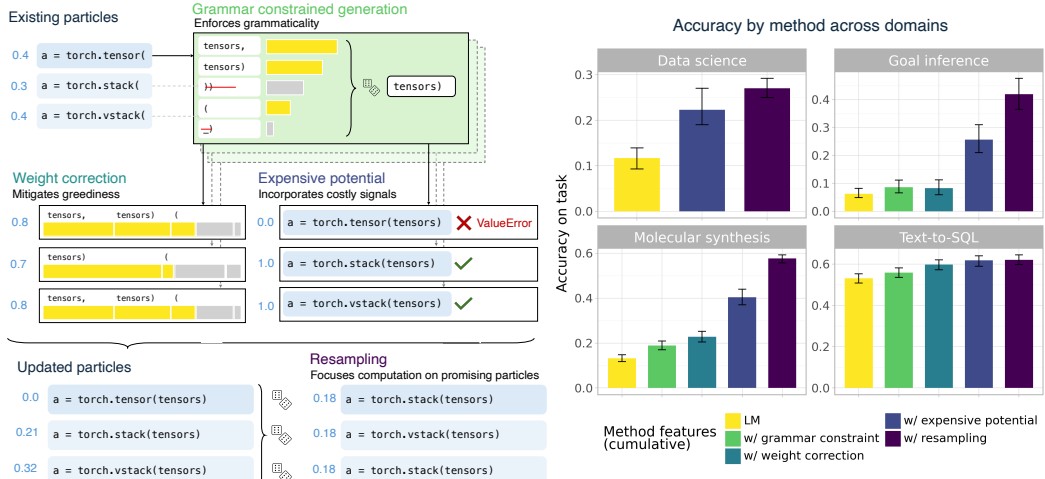

Figure 1: *Controlled generation from LMs via sequential Monte Carlo. Left:* We use sequential Monte Carlo to sample from high-quality approximations to posteriors over LM outputs. Partial sequences are repeatedly extended via grammar-constrained generation. We then apply weight corrections to mitigate the greediness of locally constrained decoding, as well as expensive potentials to encode rich information that cannot be included in logit masks. Finally, resampling focuses computation on promising particles. *Right:* Accuracy gains from these innovations on challenging data science, text-to-SQL, goal inference, and molecule synthesis benchmarks.

mitigates the greediness of locally constrained decoding; *expensive potentials*, which incorporate useful signals that several baseline methods cannot integrate; and *adaptive resampling*, which adaptively refocuses computation on partial sequences that look more promising.

- *Empirical validation of the probabilistic perspective.* We derive estimators of the KL divergence from each method's output distribution to the global product of experts (Appendix D). We find that the best-performing methods (i) have outputs that are closer in KL divergence to the global product of experts within each problem instance, and (ii) assign probabilities that are more correlated with downstream performance across problem instances (§3.3).

## 2 MONTE CARLO INFERENCE FOR CONSTRAINED SEMANTIC PARSING

**Notation.** We use $\boldsymbol{x}$ to refer to a sequence of tokens, with $x_i$ being the $i^{\text{th}}$ token in the sequence. Let $\boldsymbol{x}_{<t} \overset{\text{def}}{=} x_1 \cdots x_{t-1}$. Let $\varepsilon$ denote the empty sequence. We use juxtaposition (e.g., $\boldsymbol{x}\,\boldsymbol{y}$) to denote sequence concatenation. Let $\mathcal{A}$ be a vocabulary of tokens, and let $\mathcal{A}^*$ denote the set of all finite sequences of tokens in $\mathcal{A}$. We refer to the set $\mathcal{A}^*$ as the set of **partial sequences**. We use $\text{EOS} \notin \mathcal{A}$ to denote a special token marking the end of sequences not included in $\mathcal{A}$. We define $\mathcal{A}_{\text{EOS}} \overset{\text{def}}{=} \mathcal{A} \cup \{\text{EOS}\}$ and the set of **complete sequences** $\mathcal{A}^*\text{EOS} \overset{\text{def}}{=} \{\boldsymbol{x}\,\text{EOS} \mid \boldsymbol{x} \in \mathcal{A}^*\}$.

**Language models.** A **language model** $p$ is a probability distribution over complete sequences (i.e., $\sum_{\boldsymbol{x} \in \mathcal{A}^*\text{EOS}} p(\boldsymbol{x}) = 1$). We assume that $p$ provides a conditional distribution $p(x' \mid \boldsymbol{x})$ over its next token $x' \in \mathcal{A}_{\text{EOS}}$ given any sequence $\boldsymbol{x} \in \mathcal{A}^*$; the probability of any complete sequence factors as

$$p(\boldsymbol{x}) = \prod_{t=1}^{|\boldsymbol{x}|} p(x_t \mid \boldsymbol{x}_{<t}) \tag{2}$$

We find it convenient to extend the definition of $p(\boldsymbol{x})$ from Eq. (2) to partial sequences $\boldsymbol{x} \in \mathcal{A}^*$; note, however, that $p(\boldsymbol{x})$ is a *prefix* probability,[1] so $p$ is *not* a probability distribution over partial sequences. With this extended definition, we have $p(x' \mid \boldsymbol{x}) = \frac{p(\boldsymbol{x}\,x')}{p(\boldsymbol{x})}$ provided $p(\boldsymbol{x}) > 0$.

**Potential functions.** We consider a set $\Phi$ of domain- or task-specific **potential functions** that encode relevant constraints or preferences as nonnegative scores. Each potential function $\phi \in \Phi$ has the type $\phi \colon (\mathcal{A}^* \cup \mathcal{A}^*\text{EOS}) \to \mathbb{R}_{\geq 0}$, meaning that it assigns a non-negative real $\phi(\boldsymbol{x})$ when evaluated on some sequence $\boldsymbol{x}$, freely using any structure in the sequence so far regardless of whether $\boldsymbol{x}$

---

[1]I.e., $p(\boldsymbol{x})$ is the probability that a *complete* sequence $\boldsymbol{x}' \sim p$ has the *partial* sequence $\boldsymbol{x}$ as a prefix.

is a partial or complete sequence. For technical reasons, we assume that all potentials satisfy $\phi(\boldsymbol{x}) = 0 \implies \phi(\boldsymbol{x}\,\boldsymbol{y}) = 0$, for all $\boldsymbol{x}$ and $\boldsymbol{y}$ such that $\boldsymbol{x}\,\boldsymbol{y} \in \mathcal{A}^*\,\texttt{EOS}$.

**Target distribution.** We formalize the goal of controlled generation as sampling sequences $\boldsymbol{x} \in \mathcal{A}^*\,\texttt{EOS}$ from the **target distribution** given by the **global product of experts** between $p$ and $\Phi$:[2]

$$g(\boldsymbol{x}) = \frac{1}{Z}\,p(\boldsymbol{x})\Phi(\boldsymbol{x}) \quad \text{where} \quad Z = \sum_{\boldsymbol{y} \in \mathcal{A}^*\,\texttt{EOS}} p(\boldsymbol{y})\Phi(\boldsymbol{y}) \tag{3}$$

For intuition, if $\Phi(\boldsymbol{x}) \in \{0, 1\}$ for all $\boldsymbol{x}$ in $\mathcal{A}^*\,\texttt{EOS}$, the global product of experts can be understood as the **rejection sampling** distribution that arises by repeatedly generating $\boldsymbol{x} \sim p$, and rejecting if $\Phi(\boldsymbol{x}) = 0$. The normalizing constant $Z$ is the rate at which samples are accepted. Thus, the expected runtime of rejection sampling is $1/Z$ per accepted sample, making it expensive if $Z$ is small. Our work aims to accurately approximate the global product of experts with much less computation.

**Locally constrained decoding.** A popular approach[3] to enforcing constraints at inference-time is to apply them before sampling each token. In this approach, at each time step $t$, the current sequence $\boldsymbol{x}_{<t}$ is extended with a new token $x_t \sim \ell_\Phi(\cdot \mid \boldsymbol{x}_{<t})$ (until $x_t = \texttt{EOS}$) where[4]

$$\ell_\Phi(x_t \mid \boldsymbol{x}_{<t}) \stackrel{\text{def}}{=} \frac{p(x_t \mid \boldsymbol{x}_{<t})\frac{\Phi(\boldsymbol{x}_{<t}\,x_t)}{\Phi(\boldsymbol{x}_{<t})}}{L_\Phi(\boldsymbol{x}_{<t})} \quad \text{where} \quad L_\Phi(\boldsymbol{x}_{<t}) \stackrel{\text{def}}{=} \sum_{x' \in \mathcal{A}_{\texttt{EOS}}} p(x' \mid \boldsymbol{x}_{<t})\frac{\Phi(\boldsymbol{x}_{<t}\,x')}{\Phi(\boldsymbol{x}_{<t})} \tag{4}$$

We write $\ell_\Phi(\boldsymbol{x}) \stackrel{\text{def}}{=} \prod_{t=1}^{|\boldsymbol{x}|} \ell_\Phi(x_t \mid \boldsymbol{x}_{<t})$ for either the probability of $\boldsymbol{x} \in \mathcal{A}^*\,\texttt{EOS}$ or the prefix probability of $\boldsymbol{x} \in \mathcal{A}^*$. Note that in the former case, $\ell_\Phi$ is a distribution over complete sequences. We call it a *local* **product of experts** model because normalization is performed *locally* (at each step of the sequence), rather than *globally* (once per complete sequence).

Despite its popularity, locally constrained decoding has two important shortcomings. First, for most practical potential functions, the local and global product of experts do not define the same distribution (Lew et al., 2023; Park et al., 2025). In particular, while the global product of experts is defined with respect to complete sequences, the local product typically only has access to the string generated so far and a single token of lookahead—which can lead to myopic sampling down paths that lead to globally poor solutions. In principle, this problem can be mitigated by the choice of intermediate potentials ($\Phi(\boldsymbol{x})$ for $\boldsymbol{x} \in \mathcal{A}^*$), which implement more aggressive forms of lookahead. In particular, locally constrained decoding is an *exact* sampler when $\Phi(\boldsymbol{x}) = \Phi^*(\boldsymbol{x})$, the **expected future potential** of $\boldsymbol{x}$, $\Phi^*(\boldsymbol{x}) \stackrel{\text{def}}{=} \mathbb{E}_{\boldsymbol{x}' \sim p}\left[\Phi(\boldsymbol{x}') \mid \boldsymbol{x} \text{ is a prefix of } \boldsymbol{x}'\right]$.[5] Unfortunately, much like $Z$, computing $\Phi^*$ exactly is typically intractable. Although we may seek to approximate it, for example, by learning (Zhao et al., 2024) or adaptive methods (Park et al., 2025), here we instead focus on variants of locally constrained decoding which marginalize over the immediate next token as in Eq. (4); see Footnote 3. Our experiments (§3) compare our method to this dominant form of local decoding from the literature, using efficient tests for whether the addition of a single candidate token can satisfy the constraint.

The second, related problem with locally constrained decoding is that the local product of experts can only be sampled efficiently when it is possible to cheaply evaluate the potentials $\phi \in \Phi$ on all possible one-token continuations $\boldsymbol{x}_{<t}\,x_t'$ of the current sequence. For some constraints (e.g., checking membership or prefixhood in the language of a regular expression or context-free grammar), algorithms exist for efficient parallel evaluation across tens of thousands of possible continuations. However, for many of the constraints of interest in the present paper (including several listed in Table 1, for example, error-checking with test cases) this is not feasible. In what follows, we will assume that the set of potentials $\Phi$ can be partitioned into **expensive potentials** $\Phi_{\text{exp}}$, which are too costly to use as part of locally constrained decoding, and **efficient potentials** $\Phi_{\text{eff}}$, which can be used in sampling from the local product of experts.

**Importance sampling.** The shortcomings of local decoding can be addressed with **importance sampling**, a Monte Carlo technique for approximating intractable distributions. We describe a

---

[2]Note that care has to be taken to ensure that the sum which defines the normalizing constant in Eq. (3) converges. One sufficient (but not necessary) condition ensuring this is if $\Phi$ is bounded above by a constant.

[3]E.g., Shin et al. (2021); Scholak et al. (2021); Poesia et al. (2022); Willard & Louf (2023); Ugare et al. (2024).

[4]Here our assumption that $\phi(\boldsymbol{x}) = 0 \implies \phi(\boldsymbol{x}\,\boldsymbol{y}) = 0$ ensures that whenever $\Phi(\boldsymbol{x}_{<t}) = 0$, all extensions will be 0 as well, making it safe in such cases to define $\frac{\Phi(\boldsymbol{x}_{<t}\,x)}{\Phi(\boldsymbol{x}_{<t})} = 0$.

[5]$\Phi^*$ is also known in the SMC literature as the *optimal twist function* (e.g., Zhao et al., 2024).

particular application of the technique specialized to our setting. Here, we use the local product of experts model $\ell_{\Phi_{\text{eff}}}$ (abbreviated $\ell_{\text{eff}}$) with respect to just $\Phi_{\text{eff}}$ as a **proposal distribution**, from which we sample multiple complete **particles** $\boldsymbol{x}^{(1)}, \dots, \boldsymbol{x}^{(N)} \overset{\text{i.i.d.}}{\sim} \ell_{\text{eff}}$. For each particle $\boldsymbol{x}^{(i)}$ we define its **importance weight** $w^{(i)}$ as

$$w^{(i)} \overset{\text{def}}{=} \frac{p(\boldsymbol{x}^{(i)}) \cdot \Phi(\boldsymbol{x}^{(i)})}{\ell_{\text{eff}}(\boldsymbol{x}^{(i)})} = \left( \prod_{t=1}^{|\boldsymbol{x}^{(i)}|} L_{\text{eff}}(\boldsymbol{x}_{<t}^{(i)}) \right) \cdot \Phi_{\text{exp}}(\boldsymbol{x}^{(i)}) \tag{5}$$

The numerator is an unnormalized variant of the target distribution $g$, which we write as $\widetilde{g}$ hereafter, while the denominator $\ell_{\text{eff}}$ is the proposal distribution that was used to draw the sequence. These weighted particles define the following **posterior approximation**: $\widehat{g}(\boldsymbol{x}) \overset{\text{def}}{=} \frac{\sum_{i=1}^N w^{(i)} \mathbb{1}\{\boldsymbol{x}=\boldsymbol{x}^{(i)}\}}{\sum_{j=1}^N w^{(j)}}$, which under mild conditions converges to $g$ as $N$ grows.[6] Our importance weights simplify as shown in Eq. (5), and we note how they correct for the two problems we identified with $\ell_{\text{eff}}$. The first factor, $\prod_{t=1}^{|\boldsymbol{x}^{(i)}|} L_{\text{eff}}(\boldsymbol{x}_{<t}^{(i)})$, corrects for the greediness of $\ell_{\text{eff}}$, penalizing particles where *all* possible continuations $x_t \in \mathcal{A}_{\text{EOS}}$ score poorly in context. The second factor, $\Phi_{\text{exp}}(\boldsymbol{x}^{(i)})$, incorporates the expensive potentials that could not be used in $\ell_{\text{eff}}$. These importance weights can be computed efficiently: the first factor is already computed as a byproduct of sampling from $\ell_{\text{eff}}$, and the second factor is computed by running each of the expensive efficient potentials once on each $\boldsymbol{x}^{(i)}$.

**Sequential Monte Carlo.** While importance sampling addresses several shortcomings of local decoding, it too suffers from a major weakness: weight corrections and expensive potentials are not integrated until after a complete sequence has been generated from the proposal. This is despite the fact that critical information about whether a sequence can satisfy a constraint is often available much earlier and can be used to avoid large amounts of unnecessary computation. Sequential Monte Carlo (**SMC**; e.g., Chopin & Papaspiliopoulos, 2020), is a natural generalization of importance sampling that constructs importance-weighted samples from a *sequence* of unnormalized target distributions $\langle \widetilde{g}_t \rangle_{t=0}^\infty$ to arrive at the final unnormalized target $\widetilde{g}$. In our case, we consider intermediate targets $\widetilde{g}_t$ defined on the sequence of spaces $\mathcal{A}^{(t)} \overset{\text{def}}{=} \{\boldsymbol{x} \in \mathcal{A}^* \mid |\boldsymbol{x}| = t\} \cup \{\boldsymbol{x} \in \mathcal{A}^* \text{EOS} \mid |\boldsymbol{x}| \le t\}$, that is, partial sequences of length equal to $t$ and complete sequences of length less than or equal to $t$. The targets are defined as

$$\widetilde{g}_t(\boldsymbol{x}) = p(\boldsymbol{x})\Phi(\boldsymbol{x}), \text{ for } \boldsymbol{x} \in \mathcal{A}^{(t)} \tag{6}$$

Note that $\widetilde{g}$ and $\widetilde{g}_t$ are unnormalized distributions over different spaces: $\mathcal{A}^* \text{EOS}$ and $\mathcal{A}^{(t)}$ respectively. Whereas $\widetilde{g}$ only considers potentials on *complete* sequences, $\widetilde{g}_t$ depends also on the behavior of the potentials $\Phi_{\text{exp}}$ when applied to *partial* sequences. But as $t \to \infty$, there is less and less mass on partial sequences, and $\widetilde{g}_t$ approaches $\widetilde{g}$ no matter how the partial potentials are defined.

The particles for $\widetilde{g}_t$ are drawn as *prefixes* from $\ell_{\text{eff}}$ (stopping at length $t$ if EOS has not been reached), again requiring an importance weighting correction. The importance weight at time $t$ can be re-expressed as the importance weight from time $t-1$ times a correction factor for step $t$:

$$w_t^{(i)} \overset{\text{def}}{=} \frac{\widetilde{g}_t(\boldsymbol{x}_{<t}^{(i)} x_t^{(i)})}{\ell_{\text{eff}}(\boldsymbol{x}_{<t}^{(i)} x_t)} \tag{7a}$$

$$= \widetilde{g}_0(\boldsymbol{x}_{<1}^{(i)}) \prod_{t'=1}^t \frac{\widetilde{g}_{t'}(\boldsymbol{x}_{<t'}^{(i)} x_{t'}^{(i)})}{\widetilde{g}_{t'-1}(\boldsymbol{x}_{<t'}^{(i)}) \, \ell_{\text{eff}}(x_{t'} \mid \boldsymbol{x}_{<t'}^{(i)})} \tag{7b}$$

$$= w_{t-1}^{(i)} \frac{\widetilde{g}_t(\boldsymbol{x}_{<t}^{(i)} x_t^{(i)})}{\widetilde{g}_{t-1}(\boldsymbol{x}_{<t}^{(i)}) \, \ell_{\text{eff}}(x_t \mid \boldsymbol{x}_{<t}^{(i)})} \tag{7c}$$

$$= w_{t-1}^{(i)} \cdot L_{\text{eff}}(\boldsymbol{x}_{<t}^{(i)}) \cdot \frac{\Phi_{\text{exp}}(\boldsymbol{x}_{<t}^{(i)} x_t^{(i)})}{\Phi_{\text{exp}}(\boldsymbol{x}_{<t}^{(i)})} \tag{7d}$$

The sequential Monte Carlo algorithm generates approximations to each intermediate target $\widetilde{g}_t$, using **resampling steps** to reallocate computation from less to more promising partial sequences. We begin with a collection of $N$ weighted particles $(\boldsymbol{x}^{(i)}, w^{(i)}) = (\varepsilon, 1)$, where $\varepsilon$ is the empty

---

[6]However, the number of particles required to obtain a *good* approximation of the target distribution is exponential in the KL divergence between target $g$ and proposal $\ell_{\text{eff}}$ (Chatterjee & Diaconis, 2018).

sequence of tokens. Then, starting at $t = 1$, we repeat the following three steps until all particles are EOS-terminated (i.e., $\boldsymbol{x}^{(i)} \in \mathcal{A}^* \text{EOS}$ for all $i$):

1. *Extend.* For each incomplete particle $\boldsymbol{x}^{(i)}$, sample $x_t^{(i)} \sim \ell_{\text{eff}}(\cdot \mid \boldsymbol{x}_{<t}^{(i)})$ and update $\boldsymbol{x}^{(i)} \leftarrow \boldsymbol{x}^{(i)} \, x_t^{(i)}$.

2. *Reweight.* For each extended particle $\boldsymbol{x}^{(i)}$, update the weight $w^{(i)} \leftarrow w^{(i)} \, L_{\text{eff}}(\boldsymbol{x}_{<t}^{(i)}) \frac{\Phi_{\exp}(\boldsymbol{x}_{<t}^{(i)} x_t^{(i)})}{\Phi_{\exp}(\boldsymbol{x}_{<t}^{(i)})}$.

3. *Resample.* Sample ancestor indices $a^{(1)}, \dots, a^{(N)} \overset{\text{i.i.d.}}{\sim} \text{Categorical}\left(\frac{w^{(1)}}{W}, \dots, \frac{w^{(N)}}{W}\right)$ where $W = \sum_{i=1}^N w^{(i)}$. Then, reassign all particles $(\boldsymbol{x}^{(i)}, w^{(i)}) \leftarrow (\boldsymbol{x}^{(a^{(i)})}, \frac{W}{N})$ for all $i$ simultaneously.[7]

The extension step extends each incomplete particle with a next token proposed by the local product of experts $\ell_{\text{eff}}$. The reweighting step computes the updated importance weight, incorporating a new factor for the new token. The resampling step exploits any early signal available in the updated weights at time $t$ to abandon some less promising incomplete particles (which are unlikely to be chosen as ancestors) and focus more future computation on more promising particles (which are likely to be chosen as ancestors multiple times and then will be extended in multiple ways at time $t + 1$). This reallocation of computation often leads to dramatic improvements in inference quality—without it, SMC would reduce to the previous importance sampling method.

**Further extensions.** We further extend our SMC implementation in two ways. First, potentials in $\Phi_{\text{eff}}$ may still be modestly expensive to evaluate on the entire vocabulary. In these cases, we develop cheap stochastic approximations to the local product of distributions and use these as proposals during the *Extend* step. The incremental weight computation must also be corrected to account for these approximations; we derive stochastic unbiased estimators of the incremental weights that can be soundly used within SMC (see Appendix C). Second, the intermediate targets $\widetilde{g}_t$ do not need to advance token-by-token; in some domains, it is beneficial to consider more semantically meaningful increments. For example, in one of our experiments, the intermediate target $p_t$ is defined over the space of all partial Python programs containing $t$ or fewer lines of code (rather than tokens); the *Extend* step then samples a different number of tokens per particle, waiting in each partial sequence until a new full line has been generated. Such strategies can lead to better *particle alignment* (Lundén et al., 2018), making resampling more effective.

## 3 EXPERIMENTS

We compare seven approaches to constrained generation:

1. *Language model (Base LM).* This method simply samples from the base language model $p$ (see §3.1 for details on the language models used).
2. *Language model with grammar constraint (Locally constrained decoding).* This is the approach used by much prior work (Footnote 3). In each of our domains, we formulate a context-free grammar (CFG) encoding a notion of syntactic well-formedness appropriate for the domain (see §3.1). We let $\Phi_{\text{eff}}$ encode the binary function that determines whether its input is a prefix of some valid sequence in the grammar's language. This baseline directly samples from $\ell_{\text{eff}}$, i.e., it uses per-token logit masking to greedily enforce the CFG constraint.
3. *Language model with grammar constraint and weight correction (Grammar-only IS).* This method generates particles from $\ell_{\text{eff}}$, then computes importance weights to correct toward the global product of $p$ and $\Phi_{\text{eff}}$. These weights mitigate some of the greediness of local-product-of-experts sampling, but do not yet integrate any potentials beyond $\Phi_{\text{eff}}$.
4. *Language model with grammar constraint, weight correction, and resampling (Grammar-only SMC).* This method is a straightforward application of Lew et al. (2023) to locally constrained decoding and is similar to Park et al. (2025), which also attempts to correct for the greediness of locally constrained decoding. As in the previous method, it targets the global product of $p$ and $\Phi_{\text{eff}}$ but uses resampling to reallocate computation to promising particles.
5. *Language model with grammar constraint and expensive potential (Sample-Rerank).* Sample-Rerank is a common family of approaches for incorporating an external signal into an LM's generations post-hoc, for instance by choosing the best-of-$n$ particles via a reward model (Nakano et al., 2021; Krishna et al., 2022; Zhou et al., 2023; Gui et al., 2024; Mudgal et al., 2024; Ichihara

---

[7]In practice, we only resample if the **effective sample size** $\widehat{N} \overset{\text{def}}{=} \frac{\left(\sum_{i=1}^N w^{(i)}\right)^2}{\sum_{i=1}^N (w^{(i)})^2}$ is under a threshold (e.g., $\frac{N}{3}$).

Table 1: Summary of tasks and potential functions. Examples are truncated for brevity. Full prompts include additional information.

| Task | Potentials | | Examples | |
|---|---|---|---|---|
| | $\Phi_{\text{eff}}$ | $\Phi_{\text{exp}}$ | Prompt | Output |
| Goal Inference | STRIPS parser | Plan simulation | Write the STRIPS goal condition for the planning problem described below [...]. The STRIPS initial condition is: [...] | `(:goal (and (arm-empty) (on-table b1) [...]` |
| Python Data Science | - | Error-checking with test cases | Here is a sample dataframe: [...] I'd like to add inverses of each existing column to the dataframe [...] | `result = df.join(df.apply(lambda x: 1/x) [...]` |
| Text-to-SQL | SQL parser | Alias and table-column checking | Here is a database schema: [...] For each stadium, how many concerts are there? | `SELECT T2.name, COUNT(*) FROM concert AS T1 [...]` |
| Molecular Synthesis | SMILES parser | Incremental molecule validation | Given the following list of molecules in SMILES format, write an additional molecule [...] | `CC1=CC2(OC=N)C(=O) [...]` |

et al., 2025) or filtering via a verifier (Olausson et al., 2023; Chen et al., 2024; Lightman et al., 2024; Xin et al., 2024). In each domain, we formulate an additional potential $\Phi_{\text{exp}}$ that encodes task-specific signals of sequence quality (see §3.1). This baseline generates grammar-constrained sequences from the local product of experts, then reweights each sequence $x$ by $\Phi_{\text{exp}}(x)$.

6. *Language model with grammar constraint, weight correction, and expensive potential (Full IS).* This is the full importance sampling method described in §2, with $\Phi = \Phi_{\text{eff}} \cup \Phi_{\text{exp}}$. Unlike in the previous method, the importance weights here include correction terms that mitigate the greediness of local sampling, targeting the global product $g$. We include this method primarily as an ablation of our next method (SMC), modified not to include incremental resampling.

7. *Language model with grammar constraint, weight correction, expensive potential, and resampling (Full SMC).* This method includes all of the algorithmic contributions of our approach. It is the full sequential Monte Carlo algorithm, with $\Phi = \Phi_{\text{eff}} \cup \Phi_{\text{exp}}$. It targets the same global posterior $g$ as the previous method but uses resampling to reallocate computation to promising particles.

We report results using $N = 10$ particles; see Appendix A.2 and Fig. 2 for downstream accuracy results for a varying number of particles. We ran experiments on GCP instances with 1 A100 GPU and 12 vCPUs (our CFG parser is implemented for CPU and is parallelized across particles), with the exception of the Data Science domain, for which we used 4 H100 GPUs and 64 vCPUs.

## 3.1 DOMAINS

We study the performance of our proposed sampling methods on four challenging semantic parsing domains, summarized in Table 1; see Appendix E for further details.

- **Goal inference (Planetarium).** *Task:* Formally specify an agent's goal in the STRIPS subset of the PDDL planning language, based on a natural-language description of the goal and PDDL code detailing the agent's initial conditions and plan for achieving it. *Data:* Blocksworld tasks with up to 10 objects from the Planetarium benchmark (Zuo et al., 2024). *Metric:* Accuracy with respect to ground-truth PDDL goal. *Base LM:* Llama 3.1 8B. *Grammar:* STRIPS syntax for goals within Planetarium Blocksworld's domain definition. *Expensive potential:* Run a simulation with a ground-truth plan and check whether the resulting state conforms to the predicted (partial) goal.
- **Python for data science (DS-1000).** *Task:* Generate Python code that uses standard data science libraries (NumPy, PyTorch, Pandas, etc.) to solve a task specified in natural language and via (executable) test cases. *Data:* DS-1000 benchmark (Lai et al., 2023). *Metric:* Accuracy of the generated program with respect to the provided test cases. *Base LM:* Llama 3 70B. *Grammar:* We use a trivial potential $\Phi_{\text{eff}}(x) = 1$, as we find that the unconstrained LM reliably generates grammatical Python (that may nonetheless induce runtime errors). *Expensive potential:* Given a partial program $x$, $\Phi_{\text{exp}}$ truncates $x$ to the longest prefix of the sequence that consists of only valid Python statements (discarding any incomplete material at the end), and executes the resulting (partial) program on the provided test case, checking for runtime errors.
- **Text-to-SQL (Spider).** *Task:* Generate SQL queries from a natural language question and a database schema. *Data:* Spider development split (Yu et al., 2018). *Metric:* Execution accuracy (whether the generated SQL query, when run against a test database, produces the same results

Table 2: Comparison of method performance across domains with bootstrapped 95% confidence intervals. For brevity, *grammar constraint* and *weight correction* are abbreviated as *grammar* and *correction*, respectively.

| Method | Score | | | |
|---|---|---|---|---|
| | Goal inference | Molecular synthesis | Data science | Text-to-SQL |
| Base LM | 0.063 (0.05, 0.08) | 0.132 (0.12, 0.15) | 0.213 (0.19, 0.24) | 0.531 (0.51, 0.55) |
| *w/ grammar constraint* (Locally constrained Decoding) | 0.086 (0.07, 0.11) | 0.189 (0.17, 0.21) | - | 0.559 (0.54, 0.58) |
| *w/ grammar, weight correction* (Grammar-only IS) | 0.083 (0.06, 0.11) | 0.228 (0.21, 0.25) | - | 0.597 (0.57, 0.62) |
| *w/ grammar, potential* (Sample-Rerank) | 0.289 (0.24, 0.34) | 0.392 (0.36, 0.42) | - | 0.581 (0.56, 0.60) |
| *w/ grammar, correction, and resampling* (Grammar-only SMC) | 0.401 (0.34, 0.46) | 0.205 (0.18, 0.23) | - | 0.596 (0.57, 0.62) |
| *w/ grammar, potential, and correction* (Full IS) | 0.257 (0.21, 0.31) | 0.404 (0.37, 0.44) | 0.346 (0.31, 0.39) | **0.618** (0.59, 0.64) |
| *w/ grammar, potential, correction, and resampling* (Full SMC) | **0.419** (0.37, 0.48) | **0.577** (0.56, 0.59) | **0.407** (0.36, 0.45) | **0.620** (0.60, 0.64) |

Figure 2: **Left:** Performance on the Data Science task (DS-1000) for different models and methods. Codex-002 performance as reported in Lai et al. (2023). **Right:** Performance across all tasks for Full IS and Full SMC with 5, 10, and 50 particles. **Error bars:** bootstrapped 95% confidence intervals.

    as the ground-truth SQL query). *Base LM:* Llama 3.1 8B-Instruct. *Grammar:* SQL context-free grammars released by Roy et al. (2024), which enforce valid SQL syntax. *Expensive potential:* Check whether column names in the generated (partial) query actually belong to the queried tables, modulo aliasing. (The grammar ensures only that the column names exist in *some* table.)

- **Molecular synthesis (GDB-17).** *Task:* Generate drug-like molecules in the SMILES format (Weininger, 1988). *Data:* Few-shot prompts constructed by repeatedly choosing 20 random examples from the GDB-17 dataset (Ruddigkeit et al., 2012). *Metric:* Quantitative Estimate of Drug-likeness (QED; Bickerton et al., 2012), a standard molecular fitness function. *Base LM:* Llama 3.1 8B. *Grammar:* SMILES syntax for molecules. *Expensive potential:* A SMILES prefix validator implemented in the Python *partialsmiles* library (O'Boyle, 2024).

## 3.2 EVALUATION OF DOWNSTREAM PERFORMANCE

We begin by investigating whether our approach leads to significant performance gains. Table 2 reports posterior-weighted accuracy for our approach and ablations of its components: grammar constraints, weight corrections, expensive potentials, and resampling. We first summarize the observed effects of each component in our approach:

**Grammar constraints.** In line with previous literature (e.g., Shin et al., 2021; Scholak et al., 2021; Poesia et al., 2022; Wang et al., 2024), we find that the addition of a grammar constraint via $\Phi_{\text{eff}}$ improves downstream accuracy relative to the base LM across all domains in which it is used, even without the use of weight corrections.

**Expensive potentials.** Furthermore, we observe that integrating expensive potentials $\Phi_{\text{exp}}$ improves accuracy in models. Even without any weight corrections, the improvement in the goal inference, data science, and molecular synthesis domains is large; in the text-to-SQL domain, it is smaller but statistically significant (paired permutation test, $p < 0.01$). This suggests that making use of information that cannot be efficiently encoded in logit masks can greatly improve performance.

**Weight corrections.** Although the use of $\Phi_{\text{eff}}$ and $\Phi_{\text{exp}}$ alone leads to significant gains in downstream accuracy, these gains can be amplified with the addition of weight corrections. In cases without the expensive potential, weight corrections provide significant albeit relatively small gains in accuracy across three domains; in goal inference, it does not significantly affect performance. In the presence of the expensive potential, adding weight corrections improves accuracy for text-to-SQL and has no effect on goal inference and molecular synthesis. Overall, these results indicate that debiasing samples from a local product of experts to correctly target the global product of experts often significantly improves downstream accuracy and never harms it. That said, the accuracy gains attributable to weight corrections are modest compared to other components of the algorithm, which suggests that the bias from locally constrained decoding may be less severe in these semantic parsing domains than has been observed in other domains (e.g., constrained generation of natural language, Lew et al., 2023).

Figure 3: Estimated KL between the algorithm and the global product of experts for a representative problem instance in each domain. Values closer to 0 indicate that the algorithm is better at approximating $g$. Significant differences are indicated with ** for $p < 0.01$ and *** for $p < 0.001$ (t-test). Algorithms use $N = 10$ particles.

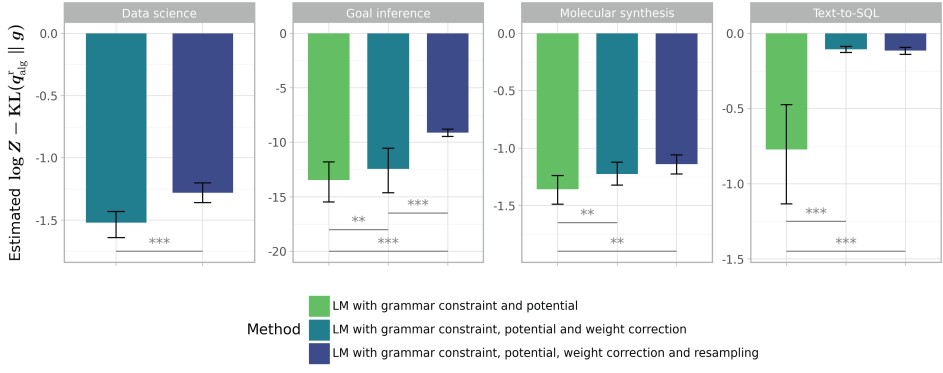

**Resampling.** We observe that the addition of resampling steps improves downstream accuracy in all domains except text-to-SQL, for which they neither significantly improve nor hurt performance. These results motivate adaptively focusing computation on promising partial sequences.

**Other Evaluations.** Next, we study the effects of varying the base language model, the number of particles used by different methods, and the computational cost of our approach: Tables 4, 6 and 8 in the appendix report the results of these experiments. We summarize key findings:

- **Our approach allows smaller LMs to outperform larger ones:** In 3 out of 4 domains (Data Science, Molecular Synthesis, Goal Inference), Full SMC allows small language models to outperform models over 8 times larger (see Tables 2 and 4). These gains persist on larger models: Fig. 2 shows how our method allows Llama 3.1 70b to outperform Codex-002, which has 175b parameters and is fine-tuned for coding tasks.
- **Our approach makes better use of resources than approaches that apply constraints only at the end of generation:** In 3 out of 4 domains (Data Science, Molecular Synthesis, Text-to-SQL), Full SMC performs as well as or better than Full IS while using one-tenth of the particles (see Fig. 2 and Table 6); in the remaining domain (Goal Inference), Full SMC outperforms IS with one fifth (10 vs 50) or one half (5 vs 10) of the particles. This is in line with the arguments drawn in §2 for the poor scaling of importance sampling and the benefits of resampling.
- **Our approach incurs minimal computational overhead:** At every token, our SMC approach incurs two computational overheads relative to a simple locally constrained decoding baseline: resampling and computing expensive potentials. Though the cost of resampling is negligible, computing expensive potentials presents a more significant cost that varies across domains: Table 8 shows that cost rarely rises above ∼30ms per token. In general, this cost is reduced by two factors: (i) expensive potentials often need to run expensive computations only at larger, semantically meaningful units (for instance, the end of a SQL clause or a Python statement) rather than at every token—therefore significantly lessening the average cost per token, (ii) expensive potentials operations are often performed on CPU rather than GPU, and therefore cost fewer dollars per hour.

### 3.3 Validation of the Probabilistic Perspective

The best-performing methods from the previous section were designed to approximate the global product of experts distribution. In this section, we investigate how closely each of these methods approximates this global distribution and whether the downstream performance results from the previous section are driven by the quality of the probabilistic inference. In particular, we find:

**Within each problem instance, the best-performing methods have outputs that are closer in KL divergence to the global product of experts.** We consider the distribution over sequences $q_{\text{alg}}^{\text{r}}(\boldsymbol{x})$ defined by each algorithm (see Appendix D for details and derivations). For each $q_{\text{alg}}^{\text{r}}$, we estimate a tractable correlate of the KL between the algorithm and the global product of experts: $\log Z - \text{KL}(q_{\text{alg}}^{\text{r}} \parallel g)$. We refer to this quantity as the *approximation quality*. Since the term $\log Z$ is algorithm-independent, we can directly compare the estimated approximation quality across algorithms to determine which ones have lower KL divergence relative to the global product of experts. However, because $\log Z$ is instance-specific, these comparisons can only be made at the instance level. Accordingly, for each domain, we select the instance with the median unique accuracy

Table 3: Pearson correlation between relative particle weights and accuracy scores for all weighted methods. Greater correlation indicates that relative weights are more strongly associated with downstream performance.

| Method | Correlation between relative weight and score | | | |
| --- | --- | --- | --- | --- |
| | Goal inference | Molecular synthesis | Data science | Text-to-SQL |
| LM with grammar constraints and weight correction (Grammar-Only IS) | 0.138 (0.10, 0.18) | 0.218 (0.16, 0.28) | 0.217 (0.18, 0.26) | 0.810 (0.79, 0.83) |
| LM with grammar constraints, potential, and weight correction (Full IS) | 0.677 (0.64, 0.71) | 0.570 (0.53, 0.61) | 0.289 (0.25, 0.33) | 0.796 (0.78, 0.81) |
| LM with grammar constraints, potential, weight correction, and resampling (Full SMC) | 0.793 (0.76, 0.82) | 0.826 (0.81, 0.84) | 0.370 (0.31, 0.42) | 0.810 (0.79, 0.83) |

Figure 4: Distributional properties of compounds generated by different methods. **Middle:** Distribution of drug-likeness as measured by QED score (Bickerton et al., 2012). **Right:** Means for other properties of interest such as diversity and de novo similarity (details in Appendix E.2).

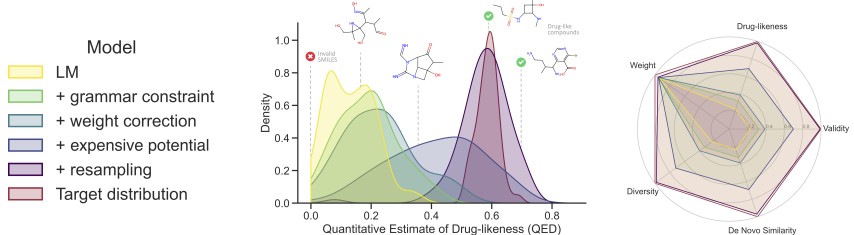

as a representative example. Fig. 3 visualizes estimated approximation quality on these examples across all methods, which include $\Phi_{\exp}$. Estimates were computed across 100 runs of each algorithm.

In all domains, sampling from the local product of experts without weight correction leads to significantly lower approximation quality relative to the methods that approximate the global product. The addition of resampling steps also significantly improves approximation quality in the data science and goal inference domains, but does not significantly change quality in the molecular synthesis and text-to-SQL domains. These trends in approximation quality are consistent with those observed in our evaluation of downstream accuracy: for example, we find that text-to-SQL is the domain in which weight corrections led to the most significant improvement in approximations of the global posterior, as well as the domain in which weight corrections most improve downstream performance. This suggests that the probabilistic formulation of the problem leads to practical gains in performance. Furthermore, these benefits can extend beyond our main performance metric; for instance, resampling during molecular generation yields simultaneous improvements along a number of additional dimensions of interest, including de-novo similarity and diversity (Fig. 4).

**Across problem instances, the best-performing methods assign probabilities that are more correlated with downstream performance** In each of our experiments, we group output particles by *semantic equivalence*, and estimate the probability of each equivalence class under the method's approximation to the global product of experts, by summing the normalized weights of the members of each equivalence class (this is similar to the postprocessing performed in Shi et al., 2022). We then measure the correlation between the estimated probability of a result and its score on the task-specific metric. Table 3 shows sequential Monte Carlo overall exhibits high correlation between (approximate) posterior probabilities and downstream performance, and that the differences in correlation between methods closely track the differences in performance in §3.2. In the goal inference, molecular synthesis, and data science domains, where expensive potentials and resampling greatly increase performance, we find that the same features also result in higher correlation between result probability and performance, whereas in text-to-SQL, where the performance gains are slimmer, we find that all methods correlate and score equally well. Together, these results validate the probabilistic approach, suggesting that the global posterior captures semantically meaningful uncertainty.

## CONCLUSION

This paper presents a principled formulation of constrained generation as sampling from a global product of experts distribution. To solve this sampling problem, we introduce a sequential Monte Carlo (SMC) algorithm that can flexibly incorporate different constraints, making incremental use of their signals. In a series of experiments, we show that this approach allows smaller models to outperform larger and fine-tuned models, that the incrementality of SMC makes it an order of magnitude more efficient than non-incremental approaches, and that downstream performance is linked to the quality of the posterior approximation, providing support for our probabilistic perspective.

ACKNOWLEDGMENTS

The authors would like to thank Manuel de Prada Corral, Brian DuSell, Joshua B. Tenenbaum, and Tan Zhi Xuan for valuable discussions, suggestions, and coding support that improved this work. The last author gratefully acknowledges the Canada CIFAR AI Chair program for support.

AUTHOR CONTRIBUTIONS

**First Authors**
- **João Loula** (`jloula@mit.edu`): research conception and development, writing, experiment development, software development (prototype)
- **Benjamin LeBrun** (`benjamin.lebrun@mail.mcgill.ca`): research conception and development, lead software engineer, experiment development, writing
- **Li Du** (`leodu@cs.jhu.edu`): experiment development, software development (parser)

**Contributors**
- **Ben Lipkin** (`lipkinb@mit.edu`): software development (grammar interfaces, testing, and integration), writing
- **Clemente Pasti** (`clemente.pasti@inf.ethz.ch`): software and algorithm development (context-free grammars), writing
- **Gabriel Grand** (`grandg@mit.edu`): software development (testing and integration), analysis and presentation of molecular synthesis experiments
- **Tianyu Liu** (`tianyu.liu@inf.ethz.ch`): software development (vLLM integration, testing)
- **Yahya Emara** (`yemara@ethz.ch`): software development (testing and integration), writing
- **Marjorie Freedman** (`mrf@isi.edu`): organization management, writing
- **Jason Eisner** (`jason@cs.jhu.edu`): technical advice, writing, project advising and mentorship
- **Ryan Cotterell** (`ryan.cotterell@inf.ethz.ch`): organization management, senior project leadership, research conception and development, writing

**Senior Authors**
- **Vikash Mansinghka** (`vkm@mit.edu`): organization management, research conception and development, project advising and mentorship
- **Alexander K. Lew** (`alexander.lew@yale.edu`): senior project leadership, research conception and development, project narrative development, writing, software development (prototype)
- **Tim Vieira** (`tim.f.vieira@gmail.com`): senior project leadership, full-stack software contributor, research conception and development, project narrative development, writing, software system design and implementation
- **Timothy J. O'Donnell** (`timothy.odonnell@mcgill.ca`): overall team leadership and direction, organization management, senior project leadership, research conception and development, project advising and mentorship, project narrative development, writing

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

Table 4: Downstream accuracy of different methods with a **smaller** base language model (Llama 3.1 8B in Data science and Llama 3.2 1B in all other domains). Errors are bootstrapped 95% confidence intervals. Instruct model is used for Text-to-SQL.

| Method | Score | | | |
|---|---|---|---|---|
| | **Goal inference** | **Molecular synthesis** | **Data science** | **Text-to-SQL** |
| LM | 0.012 (0.01, 0.02) | 0.032 (0.02, 0.04) | 0.114 (0.09, 0.14) | 0.224 (0.207, 0.241) |
| *w/ grammar constraint* (Locally constrained Decoding) | 0.046 (0.03, 0.06) | 0.031 (0.02, 0.04) | - | 0.250 (0.232, 0.270) |
| *w/ grammar, weight correction* (Grammar-only IS) | 0.037 (0.02, 0.06) | 0.041 (0.03, 0.05) | - | 0.301 (0.281, 0.323) |
| *w/ grammar, potential* (Sample-Rerank) | 0.087 (0.06, 0.12) | 0.119 (0.09, 0.16) | - | 0.299 (0.278, 0.321) |
| *w/ grammar, correction, and resampling* (Grammar-only SMC) | 0.052 (0.03, 0.08) | 0.050 (0.04, 0.06) | - | 0.302 (0.281, 0.324) |
| *w/ grammar, potential, and correction* (IS) | 0.079 (0.05, 0.11) | 0.122 (0.09, 0.16) | 0.225 (0.19, 0.26) | **0.348** (0.326, 0.372) |
| *w/ grammar, potential, correction, and resampling* (SMC) | **0.125** (0.09, 0.16) | **0.517** (0.48, 0.55) | **0.285** (0.24, 0.34) | **0.348** (0.325, 0.374) |

Table 5: Downstream accuracy of different methods with a **larger** (relative to Table 4) base language models that were used in the main experiments (Llama 3.1 70B in Data science and Llama 3.1 8B in all other domains). Errors are bootstrapped 95% confidence intervals. Instruct model is used for Text-to-SQL. This table is identical to Table 2 in the main text and is repeated in the appendix for easier comparison.

| Method | Score | | | |
|---|---|---|---|---|
| | **Goal inference** | **Molecular synthesis** | **Data science** | **Text-to-SQL** |
| LM | 0.063 (0.05, 0.08) | 0.132 (0.12, 0.15) | 0.213 (0.19, 0.24) | 0.531 (0.51, 0.55) |
| *w/ grammar constraint* (Locally constrained Decoding) | 0.086 (0.07, 0.11) | 0.189 (0.17, 0.21) | - | 0.559 (0.54, 0.58) |
| *w/ grammar, weight correction* (Grammar-only IS) | 0.083 (0.06, 0.11) | 0.228 (0.21, 0.25) | - | 0.597 (0.57, 0.62) |
| *w/ grammar, potential* (Sample-Rerank) | 0.289 (0.24, 0.34) | 0.392 (0.36, 0.42) | - | 0.581 (0.56, 0.60) |
| *w/ grammar, correction, and resampling* (Grammar-only SMC) | 0.401 (0.34, 0.46) | 0.205 (0.18, 0.23) | - | 0.596 (0.57, 0.62) |
| *w/ grammar, potential, and correction* (Full IS) | 0.257 (0.21, 0.31) | 0.404 (0.37, 0.44) | 0.346 (0.31, 0.39) | **0.618** (0.59, 0.64) |
| *w/ grammar, potential, correction, and resampling* (Full SMC) | **0.419** (0.37, 0.48) | **0.577** (0.56, 0.59) | **0.407** (0.36, 0.45) | **0.620** (0.60, 0.64) |

# A  ADDITIONAL EXPERIMENTS

## A.1  SMALLER BASE LMS

This section evaluates downstream accuracy across methods using smaller base language models (relative to Table 2 in the main text, reproduced in Appendix Table 5 for easier comparison). For the Text-to-SQL, Molecular Synthesis, and Goal Inference domains, which in the §3.2 experiments used Llama 3.1 (8B), we substitute Llama 3.2 (1B). In the Data Science domain, which used Llama 3 (70B) in the §3.2 experiments, we substitute Llama 3.1 (8B). All experiments were run with $N = 10$ particles, and the instruct version of Llama 3.2 (1B) was used in the text-to-SQL domain to remain consistent with the model variants used in the main paper.

We report posterior-weighted accuracy using the smaller LMs across all methods and domains in Table 4. Although accuracy is significantly lower compared to the larger LMs, we find that weight corrections, expensive potentials, and resampling steps still improve model performance. We also find that, in general, the relative gains in accuracy provided by our method are more pronounced for smaller language models. For easier comparison, Table 5 presents an identical version of Table 2, showing the results for the larger base LMs which were reported in §3.2. With the exception of Text-to-SQL, we observe that our approach with the smaller LM outperforms the locally constrained decoding baseline (LM w/ grammar constraint) using the larger LM. In the Data Science domain, our Full SMC approach with the smaller LM outperforms the larger base LM. These results suggest that our approach can dramatically improve the performance of smaller LMs.

## A.2  ACCURACY BY NUMBER OF PARTICLES

This section investigates how performance improvements vary with the number of particles. Table 6 reports downstream accuracy for $N = 5$, $N = 10$, and $N = 50$ particles using the Llama 3.1 (8B) models. Note that we only include methods in which samples are generated from an approximation that is constructed from a set of importance-weighted particles. For the base LM and locally constrained decoding baselines, samples are generated through direct ancestral sampling. As a result, the number of particles does not influence accuracy in these cases (though additional particles can provide a better estimate of the true model accuracy), so we omit these methods from the analysis.

The main effect we observe is the more efficient use of computational resources by Full SMC compared to methods that do not incorporate incremental information, such as Full IS: the former outperforms the latter with one tenth of the particles in 3 out of 4 domains (Data Science, Molecular Sythesis, Text-to-SQL) and one fifth of the particles in the other domain (Goal Inference, see Fig. 2 in the main text for a visualization). We note an additional patterns of results: in the Text-to-SQL and

Table 6: Accuracy by number of particles across methods. Errors are bootstrapped 95% confidence intervals. Llama 3.1 8B is used as the base LM for all domains. Instruct model is used for Text-to-SQL.

| Method | Score | | | |
|---|---|---|---|---|
| | Goal inference | Molecular synthesis | Data science | Text-to-SQL |
| **5 Particles** | | | | |
| *LM w/ grammar constraint, correction* (Grammar-only IS) | 0.106 (0.08, 0.14) | 0.239 (0.21, 0.27) | - | 0.587 (0.56, 0.61) |
| *LM w/ grammar constraint, potential* (Sample-Rerank) | 0.214 (0.17, 0.26) | 0.407 (0.36, 0.45) | - | 0.578 (0.55, 0.60) |
| *LM w/ grammar constraint, correction, and resampling* (Grammar-only SMC) | 0.310 (0.26, 0.37) | 0.209 (0.18, 0.24) | - | 0.599 (0.57, 0.62) |
| *LM w/ grammar constraint, potential, and correction* (Full IS) | 0.216 (0.17, 0.27) | 0.411 (0.37, 0.45) | 0.204 (0.16, 0.25) | **0.611** (0.59, 0.63) |
| *LM w/ grammar constraint, potential, correction, and resampling* (Full SMC) | **0.319** (0.27, 0.37) | **0.552** (0.52, 0.58) | **0.224** (0.18, 0.27) | **0.620** (0.59, 0.64) |
| **10 Particles** | | | | |
| *LM w/ grammar constraint, weight correction* (Grammar-only IS) | 0.083 (0.06, 0.11) | 0.228 (0.21, 0.25) | - | 0.597 (0.57, 0.62) |
| *LM w/ grammar constraint, potential* (Sample-Rerank) | 0.289 (0.24, 0.34) | 0.392 (0.36, 0.42) | - | 0.581 (0.56, 0.60) |
| *LM w/ grammar constraint, correction, and resampling* (Grammar-only SMC) | 0.401 (0.34, 0.46) | 0.205 (0.18, 0.23) | - | 0.596 (0.57, 0.62) |
| *LM w/ grammar constraint, potential, and correction* (Full IS) | 0.257 (0.21, 0.31) | 0.404 (0.37, 0.44) | 0.223 (0.19, 0.27) | **0.618** (0.59, 0.64) |
| *LM w/ grammar constraint, potential, correction, and resampling* (Full SMC) | **0.419** (0.37, 0.48) | **0.577** (0.56, 0.59) | **0.285** (0.26, 0.32) | **0.620** (0.60, 0.64) |
| **50 Particles** | | | | |
| *LM w/ grammar constraint, correction* (Grammar-only IS) | 0.069 (0.05, 0.09) | 0.211 (0.20, 0.22) | - | 0.603 (0.58, 0.63) |
| *LM w/ grammar constraint, potential* (Sample-Rerank) | 0.416 (0.36, 0.47) | 0.382 (0.37, 0.40) | - | 0.585 (0.56, 0.61) |
| *LM w/ grammar constraint, correction, and resampling* (Grammar-only SMC) | 0.595 (0.54, 0.65) | 0.212 (0.20, 0.23) | - | 0.599 (0.58, 0.62) |
| *LM w/ grammar constraint, potential, and correction* (Full IS) | 0.393 (0.35, 0.45) | 0.389 (0.38, 0.40) | 0.218 (0.19, 0.25) | **0.626** (0.60, 0.66) |
| *LM w/ grammar constraint, potential, correction, and resampling* (Full SMC) | **0.611** (0.56, 0.66) | **0.569** (0.56, 0.58) | **0.292** (0.25, 0.33) | **0.622** (0.60, 0.65) |

Table 7: Downstream accuracy comparison with the SMC Steering method from Lew et al. (2023) in the text-to-SQL domain. Errors are bootstrapped 95% confidence intervals. Both methods include expensive potentials. Our method is run with 10 particles. SMC Steering is run with 5 particles and a beam size of 3. Both methods are run with Llama 3.1 8B Instruct.

| Method | Score |
|---|---|
| Full SMC | **0.620** (0.60, 0.64) |
| SMC Steering (Lew et al., 2023) | 0.607 (0.58, 0.63) |

Molecular synthesis domains, increasing the number of particles has a marginal impact on downstream accuracy; however, in Goal inference and Data Science, we observe that a greater number of particles can lead to significantly better downstream accuracy (though only when increasing from 5 to 10 particles in the Data Science domain). Given that Goal Inference and Data Science are the two tasks where our expensive potentials are most informative, this pattern of results seems to be reflective of the fact that richer potentials require more computation to fully exploit.

## A.3 RESAMPLING WITHOUT REPLACEMENT (LEW ET AL., 2023)

This section evaluates our approach using the without-replacement resampling method introduced in Lew et al. (2023). Specifically, we use our Full SMC algorithm with expensive potential (LM w/ grammar constraint, potential, correction, and resampling), and replace multinomial resampling steps with Lew et al. (2023)'s without replacement scheme. For comparison, we ran the without replacement baseline (SMC Steering) with $N = 5$ particles and a beam size of 3, alongside our approach using multinomial resampling with $N = 10$ particles (and an ESS threshold of 0.9). These settings effectively give the SMC Steering method a particle count of $N = 15$, giving it an advantage in the comparison.

Table 7 reports weighted accuracy for these methods in the text-to-SQL domain (we restricted this analysis to a single domain because of limitations in computational resources). We observe that without-replacement resampling steps slightly hurt performance compared to multinomial resampling.

## A.4 COMPUTATIONAL COST

Though we have shown that practitioners can improve over locally constrained decoding by using our proposed SMC method, in practice, there is additional computational cost stemming from two sources: resampling and computing expensive potentials $\Phi_{\exp}$. The cost of resampling is negligible, consisting only of simple sum, softmax, and categorical sampling operations at every token. The cost of computing expensive potentials, on the other hand, is more significant and varies across domains. Table 8 shows the average per token cost of computing expensive potentials for all of our domains: we see that it rarely goes above about 30ms.

Table 8: Average per token cost (in seconds) of computing the expensive potential $\Phi_{exp}$ for each of our domains. Intervals are bootstrapped confidences estimated by selecting 10 SMC generations at random for each domain.

| Method | Goal Inference | Molecular Synthesis | Data Science | Text-to-SQL |
|---|---|---|---|---|
| $\Phi_{exp}$ seconds per token | 0.011 (0.007, 0.016) | 0.0003 (0.0002, 0.0004) | 0.007 (0.0009, 0.023) | 0.031 (0.0204, 0.0413) |

In general, the computational cost of expensive potentials is lessened by two factors: 1) expensive potentials often change not at every token, but only at larger, semantically meaningful units (for instance, the end of a SQL clause or a Python statement)—caching can therefore significantly lessen computational cost, 2) expensive potentials are often CPU rather than GPU computations (and so the cost of computation is much cheaper).

## B    GRAMMAR-BASED POTENTIALS AND PARSING ALGORITHMS

We briefly describe how we implemented the parser, i.e., grammar-based potential functions, used in our experiments (§3). These potentials are derived from a *context-free grammar* (CFG; see, e.g., Hopcroft & Ullman, 1979), a rule-based formalism for assigning non-negative values to strings. In the setting of our experiments, these are $\{0, 1\}$-valued judgments of syntactic validity, e.g., whether or not a string is a well-formed SQL query. We note, however, that context-free grammars can be easily generalized to the more general case of assigning nonnegative scores to these sequences (Goodman, 1999), not just boolean values, even though we do not leverage this flexibility in our experiments.

A CFG $\mathcal{G}$ defines a boolean-valued function $\mathcal{G} \colon \Sigma^*\texttt{EOS} \to \{0, 1\}$, where $\Sigma$ is an alphabet of terminal symbols. We call $L(\mathcal{G}) = \text{supp}(\mathcal{G})$ the *language* of $\mathcal{G}$, where $\text{supp}(\mathcal{G})$ denotes the support of $\mathcal{G}$. Intuitively, we can define a *grammar-based potential* $\phi_{\mathcal{G}}$ as a function that evaluates whether a string $\boldsymbol{y}$ is a prefix of some string in $L(\mathcal{G})$. Formally we write this as $\phi_{\mathcal{G}}(\boldsymbol{y}) = \mathbb{1}\{\exists \boldsymbol{y}' \in \Sigma^*\texttt{EOS} \mid \boldsymbol{y} \preceq \boldsymbol{y}', \boldsymbol{y}' \in L(\mathcal{G})\}$, for every $\boldsymbol{y} \in \Sigma^* \cup \Sigma^*\texttt{EOS}$, where $\boldsymbol{y} \preceq \boldsymbol{y}'$ means that $\boldsymbol{y}$ is a prefix of $\boldsymbol{y}'$. In this work, we assume that the terminal symbols are characters—or, more precisely, bytes[8]—so that they may be aligned to the token vocabulary $\mathcal{A}$ used by the language model $p$.[9] This approach assumes that each token $x \in \mathcal{A}$ is equivalent to a sequence of bytes.[10]

We implemented a specialized algorithm[11] to efficiently infer grammar-based potentials $\phi_{\mathcal{G}}(\boldsymbol{y}\,y')$ for all terminals $y' \in \Sigma \cup \{\texttt{EOS}\}$ (simultaneously) given a prefix $\boldsymbol{y} \in \Sigma^*$. Note that in principle, we could transform the potential function to have the type $(\mathcal{A}^* \cup \mathcal{A}^*\texttt{EOS}) \to \{0, 1\}$ by scoring each token $y \in \mathcal{A}_{\texttt{EOS}}$ based on its decoded byte sequence; however, the number of tokens is very large, making too expensive to evaluate the potential for each token's byte sequence, in practice. In Appendix C.2, we describe an efficient strategy for (approximately) sampling tokens the locally constrained proposal distribution (Eq. (4)), which comes with some useful probabilistic guarantees.

---

[8]This may be achieved by adding additional rules to the CFG that expand each of its terminal symbols into its constituent byte sequence.

[9]An alternative approach is to transform a byte-level CFG into a token-level CFG; however, this can make the grammar extremely large.

[10]In practice, this may require a lookup table to convert each token identifier to its representation as a byte string.

[11]Our algorithm is based on Earley (1968); Stolcke (1995); Nowak & Cotterell (2023); Opedal et al. (2023).

## C  THE SET-BASED PROPOSAL SPEEDUP

This section describes a set-based proposal speedup (Appendix C.1). We instantiate the speedup in Appendix C.2 for the special case of character-level potential functions, such as parsing.

### C.1  FRAMEWORK

The *reweight* step of our sequential Monte Carlo algorithm requires us to evaluate $L_{\mathrm{eff}}(\boldsymbol{x})$, and the *extend* step requires us to sample from the locally constrained distribution $\ell_{\mathrm{eff}}(\cdot \mid \boldsymbol{x})$. Both of these operations can be moderately expensive because they require the evaluation of $\Phi_{\mathrm{eff}}(\boldsymbol{x}\,x')$ for all tokens $x' \in \mathcal{A}_{\mathsf{EOS}}$. In this section, we describe a general scheme for speeding up both steps by *approximately* sampling $\ell_{\mathrm{eff}}(\cdot \mid \boldsymbol{x})$ and *approximately* evaluating $L_{\mathrm{eff}}(\boldsymbol{x})$. If we are careful about how we carry out this approximation, it will not change the intermediate targets $\widetilde{g}_t(\boldsymbol{x})$, defined Eq. (6), that control the behavior of the sequential Monte Carlo algorithm.

**Sequential Monte Carlo with approximate *Extend* and *Reweight* steps.** The key invariant maintained by sequential Monte Carlo is that at each step, the distribution of each particle $(\boldsymbol{x}_t^{(i)}, w_t^{(i)})$ is *properly weighted* for the intermediate target $\widetilde{g}_t$.

**Definition 1.** *Let $\tilde{p}(x) = Z_p \cdot p(x)$ be an unnormalized distribution on a space $X$, with normalizing constant $Z_p$. Let $q$ be a probability distribution on weighted pairs $(x, w) \in X \times \mathbb{R}_{\geq 0}$. We say that $q$ is properly weighted for $\tilde{p}$ if, for any function $f$,*

$$\mathop{\mathbb{E}}_{(x,w)\sim q}[w \cdot f(x)] = Z_p \mathop{\mathbb{E}}_{x\sim p}[f(x)] \tag{8}$$

The *extend* and *reweight* steps of the algorithm, which update a previous particle $(\boldsymbol{x}_{<t}^{(i)}, w_{t-1}^{(i)})$ by sampling $x_t^{(i)} \sim \ell_{\mathrm{eff}}(\boldsymbol{x}_{<t}^{(i)})$ and returning the new particle

$$\left( \boldsymbol{x}_{<t}^{(i)}\, x_t^{(i)}, w_{t-1}^{(i)} \cdot L_{\mathrm{eff}}(\boldsymbol{x}_{<t}^{(i)}) \cdot \frac{\Phi_{\exp}(\boldsymbol{x}_{<t}^{(i)}\, x_t^{(i)})}{\Phi_{\exp}(\boldsymbol{x}_{<t}^{(i)})} \right)$$

are justified by the fact that if $(\boldsymbol{x}_{<t}, w_{t-1}^{(i)})$ is properly weighted for $\widetilde{g}_{t-1}$, then the new pair is properly weighted for $\widetilde{g}_t$. We are looking to improve runtime performance without compromising soundness, so we seek ways of modifying the *extend* and *reweight* steps that do not break this invariant.

In particular, suppose that instead of sampling $x_t^{(i)} \sim \ell_{\mathrm{eff}}(\cdot \mid \boldsymbol{x}_{<t}^{(i)})$ and computing the *exact* weight update $w_t^{(i)} \stackrel{\text{def}}{=} w_{t-1}^{(i)} \cdot L_{\mathrm{eff}}(\boldsymbol{x}_{<t}^{(i)}) \cdot \frac{\Phi_{\exp}(\boldsymbol{x}_{<t}^{(i)}\, x_t^{(i)})}{\Phi_{\exp}(\boldsymbol{x}_{<t}^{(i)})}$, we instead generate $(x, W)$ from a proposal $q$ that is *properly weighted* for the unnormalized local product of experts $\widetilde{\ell}(x) \stackrel{\text{def}}{=} L_{\mathrm{eff}}(\boldsymbol{x}_{<t}^{(i)}) \cdot \ell_{\mathrm{eff}}(x \mid \boldsymbol{x}_{<t}^{(i)})$, then compute the alternative weight update

$$w_t^{(i)} \stackrel{\text{def}}{=} w_{t-1}^{(i)} \cdot W \cdot \frac{\Phi_{\exp}(\boldsymbol{x}_{<t}^{(i)}\, x)}{\Phi_{\exp}(\boldsymbol{x}_{<t}^{(i)})} \tag{9}$$

The key observation is that, by the fact that $q$ is properly weighted for $\widetilde{\ell}$, we know that for every possible previous particle $(\boldsymbol{x}_{<t}^{(i)}, w_{t-1}^{(i)})$ and every function $f$,

$$\mathop{\mathbb{E}}_{(x,W)\sim q}[w_t^{(i)} \cdot f(\boldsymbol{x}_{<t}^{(i)}\, x)] = \mathop{\mathbb{E}}_{(x,W)\sim q}\left[ w_{t-1}^{(i)} \cdot W \cdot \frac{\Phi_{\exp}(\boldsymbol{x}_{<t}^{(i)}\, x)}{\Phi_{\exp}(\boldsymbol{x}_{<t}^{(i)})} \cdot f(\boldsymbol{x}_{<t}^{(i)}\, x) \right] \tag{10}$$

$$= L_{\mathrm{eff}}(\boldsymbol{x}_{<t}^{(i)}) \mathop{\mathbb{E}}_{x_t^{(i)}\sim \ell_{\mathrm{eff}}(\cdot|\boldsymbol{x}_{<t}^{(i)})}\left[ w_{t-1}^{(i)} \frac{\Phi_{\exp}(\boldsymbol{x}_{<t}^{(i)}\, x_t^{(i)})}{\Phi_{\exp}(\boldsymbol{x}_{<t}^{(i)})} f(\boldsymbol{x}_{<t}^{(i)}\, x_t^{(i)}) \right] \tag{11}$$

$$= \mathop{\mathbb{E}}_{x_t^{(i)}\sim \ell_{\mathrm{eff}}(\cdot|\boldsymbol{x}_{<t}^{(i)})}\left[ w_t^{(i)} \cdot f(\boldsymbol{x}_{<t}^{(i)}\, x_t^{(i)}) \right] \tag{12}$$

Therefore, if the overall proper weighting invariant (with respect to the intermediate target $\widetilde{g}_t$) holds for the original update, then it will also hold for this modified *extend*-and-*reweight* procedure. For more on SMC with estimated weights, see Chopin & Papaspiliopoulos (2020) and Lew et al. (2022).

**A family of properly weighted updates based on the Horvitz–Thompson estimator.** We now introduce a useful family of properly weighted proposals for our setting. It will allow us to generate

weighted next-token proposals $(x, W)$ while *only* evaluating the potentials $\Phi_{\text{eff}}$ on a (randomly chosen) subset $S \subseteq \mathcal{A}_{\text{EOS}}$. Our procedure is inspired by the Horvitz–Thompson estimator (Horvitz & Thompson, 1952).

First, to reduce notational clutter, we define the following shorthand: $L \stackrel{\text{def}}{=} L_{\text{eff}}(\boldsymbol{x})$, and $\ell(x') = \widetilde{\ell}(x')/L = \ell_{\text{eff}}(x' \mid \boldsymbol{x})$.

**Definition 2.** *Given a probability distribution $q$ over subsets of $\mathcal{A}_{\text{EOS}}$, we define the **set-based proposal speedup** by the following generation procedure:*

1. *Sample a subset $S \sim q$ where $q$ is a probability distribution over subsets of $\mathcal{A}_{\text{EOS}}$.*
2. *Compute the* local *weight $w(x) \stackrel{\text{def}}{=} \frac{\widetilde{\ell}(x)}{\pi(x)}$ of each token $x \in S$ where $\pi_q(x)$ is the **inclusion probability** $\pi_q(x) \stackrel{\text{def}}{=} \Pr_{S' \sim q}[x \in S'] = \sum_{S'} q(S') \mathbb{1}\{x \in S'\}$, i.e., the probability that $x$ ends up in any sampled $S' \sim q$.[12]*
3. *Compute the set-conditioned distribution $q(x \mid S) \stackrel{\text{def}}{=} \frac{w(x)\,\mathbb{1}\{x \in S\}}{W_S}$ where $W_S = \sum_{x \in S} w(x)$.*
4. *Sample $x \sim q(\cdot \mid S)$.*
5. *Return $(x, W_S)$*

Then, as described above, we modify the SMC algorithm to use the sampled token $x$ instead of $x \sim \ell_{\text{eff}}(\cdot \mid \boldsymbol{x})$ in the *extend* step, and the weight $W_S$ instead of $L_{\text{eff}}(\boldsymbol{x})$ in the *reweight* step.

We justify this approach by showing that it produces properly weighted samples.

**Proposition 1.** *The set-based proposal speedup's distribution $q$ (Def. 2) is a properly weighted proposal for $\widetilde{\ell}$.*

*Proof.* Let $f$ be an arbitrary real-valued function.

$$\mathbb{E}_{(x, W_S) \sim q}[f(x) W_S] = \sum_x f(x) \sum_S q(x \mid S) q(S) W_S \tag{13a}$$

$$= \sum_x f(x) \sum_S \frac{w(x)\,\mathbb{1}\{x \in S\}}{W_S} q(S) W_S \tag{13b}$$

$$= \sum_x f(x) w(x) \sum_S \mathbb{1}\{x \in S\}\, q(S) \tag{13c}$$

$$= \sum_x f(x) \frac{\widetilde{\ell}(x)}{\pi_q(x)} \pi_q(x) \tag{13d}$$

$$= L \sum_x f(x) \ell(x) \tag{13e}$$

$$= L \,\mathbb{E}_{x \sim \ell}[f(x)] \tag{13f}$$

$\square$

### C.2 CHARACTER-BASED PROPOSAL

Our character-based proposal distribution is an instance of the framework of the previous section. In particular, $q$ samples sets of tokens $S$ by sampling a sequence of characters. We provide the pseudocode for this algorithm in Alg. 1, and define two key data structures used by this proposal:

**Definition 3.** *Our **trie data structure** $T$ is a labeled, tree-structured graph that is defined as follows:*

- *Let $\mathcal{A}_{\text{EOS}}$ be the LM's vocabulary of tokens where we represent each token as its strings of characters ending with a designated end-of-token marker EOT.[13] Let $\Sigma$ denote the set of characters (or bytes).*
- *Let $P$ be the prefix closure of the set $\mathcal{A}_{\text{EOS}}$: $P \stackrel{\text{def}}{=} \{\boldsymbol{p} \in \Sigma^* \mid \boldsymbol{p} \preceq x, x \in \mathcal{A}_{\text{EOS}}\}$ where $\boldsymbol{p} \preceq x$ denotes that $\boldsymbol{p}$ is a prefix of $x$.*

---

[12] We require $q$ to be such that every token has a positive inclusion probability $\pi_q(x) > 0$ for all $x \in \mathcal{A}_{\text{EOS}}$.

[13] Note that EOS is handled specially as it is not a string of characters.

- *Let $T = (N, E)$ be a labeled graph with node P and labeled edges $E = \left\{ \boldsymbol{p} \xrightarrow{a} \boldsymbol{p}\, a \,\middle|\, \boldsymbol{p}, (\boldsymbol{p}\, a) \in P \right\}$*

**Definition 4.** *Let* mass *be a mapping $N \to [0, 1]$, defined as follows.*

$$\texttt{mass}(x') = p(x' \mid \boldsymbol{x}), \qquad for\ x' \in \mathcal{A}_{\mathsf{EOS}} \tag{14a}$$

$$\texttt{mass}(\boldsymbol{p}) = \sum_{\boldsymbol{p} \xrightarrow{a} \boldsymbol{p}\, a \in E} \texttt{mass}(\boldsymbol{p}\, a), \qquad for\ \boldsymbol{p} \in P \setminus \mathcal{A}_{\mathsf{EOS}} \tag{14b}$$

---

**Algorithm 1** Character proposal: This procedure implements a properly weighted proposal distribution for the unnormalized version of the locally constrained distribution $\ell_{\{\phi_{\mathcal{G}}\}}(\cdot \mid \boldsymbol{x})$.

---

1. **procedure** character_proposal($\boldsymbol{x}$)
2.   mass $\leftarrow$ Apply Eq. (14) to $p(\cdot \mid \boldsymbol{x})$
3.   $\boldsymbol{p} \leftarrow \varepsilon$                  ▷ *start at the trie's root node*
4.   $c \leftarrow 1$                  ▷ *path weight under cfg*
5.   $w \leftarrow \{\}$
6.   $S \leftarrow \emptyset$
7.   $\pi \leftarrow \{\}$
8.   $\pi(\varepsilon) \leftarrow 1$
9.   **while** true :
10.     $p_1 \leftarrow \left\{ a : \frac{\texttt{mass}(\boldsymbol{p}\, a)}{\texttt{mass}(\boldsymbol{p})} \text{ for } \boldsymbol{p} \xrightarrow{a} \boldsymbol{p}\, a \in E \right\}$
11.     $p_2 \leftarrow \left\{ a : \frac{\phi_{\mathcal{G}}(\boldsymbol{x}\, \boldsymbol{p}\, a)}{\phi_{\mathcal{G}}(\boldsymbol{x}\, \boldsymbol{p})} \text{ for } a \in \Sigma \right\}$
12.     **if** $\boldsymbol{p} \xrightarrow{\texttt{EOT}} \boldsymbol{p}\,\texttt{EOT} \in E$:        ▷ *End-of-token available (i.e., $\boldsymbol{p} \in \mathcal{A}_{\mathsf{EOS}}$)*
13.      $w(\boldsymbol{p}) = \frac{\texttt{mass}(\boldsymbol{p}\,\texttt{EOT}) \cdot c}{\pi(\boldsymbol{p})}$
14.      $S \leftarrow S \cup \{\boldsymbol{p}\}$
15.     $\overline{q} \leftarrow \{a : p_1(a) \cdot p_2(a) \text{ for } a \in \Sigma\}$      ▷ *Note: $\texttt{EOT} \notin \Sigma$.*
16.     $Q \leftarrow \sum_{a \in \Sigma} \overline{q}(a)$
17.     **if** $Q = 0$:              ▷ *cannot continue further*
18.      **break**
19.     $a \sim \overline{q}/Q$           ▷ *Sample next character proportional to $\overline{q}$*
20.     $\pi(\boldsymbol{p}\, a) \leftarrow \pi(\boldsymbol{p}) \cdot \overline{q}(a)/Q$
21.     $\boldsymbol{p} \leftarrow \boldsymbol{p}\, a$        ▷ *extend the prefix (i.e., transition to the next node)*
22.     $c \leftarrow c \cdot p_2(a)$
23.   $W \leftarrow \sum_{x' \in S} w(x')$
24.   $x \sim w(\cdot)/W$
25.   **return** $(x, W)$

---

It is straightforward to verify that the character proposal is an instance of set-based proposal speedup (Def. 2), which is properly weighted according to Proposition 1.

| Name | Proposal | Target | Inference alg. |
|---|---|---|---|
| Language model | $p$ | $p$ | Exact |
| *w/ grammar constraint* | $\ell_{\text{eff}}$ | $\ell_{\text{eff}}$ | Exact |
| *w/ grammar constraint, weight correction* | $\ell_{\text{eff}}$ | $g_{\text{eff}}$ | IS |
| *w/ grammar constraint, potential* | $\ell_{\text{eff}}$ | $\ell_{\text{eff}} \cdot \Phi_{\text{exp}}$ | IS |
| *w/ grammar constraint, potential, and correction* | $\ell_{\text{eff}}$ | $g$ | IS |
| *w/ grammar constraint, potential, correction, and resampling* | $\ell_{\text{eff}}$ | $g$ | SMC |

Table 9: Potentials, models, and inference algorithms. Note that $g_{\text{eff}}$ differs from $g$, as the former only incorporates the product of efficient potentials $\Phi_{\text{eff}}$. Consistent with our approach throughout the paper, we also assume that the local product of experts $\ell_{\text{eff}}$, only features the product with efficient potentials $\Phi_{\text{eff}}$.

## D ESTIMATING INFERENCE QUALITY

The KL divergence $\text{KL}(q \parallel g)$ measures how well the distribution $q$ approximates the global product of experts $g$. The exact $\text{KL}(q \parallel g)$ is generally intractable, for two reasons: $g$ is known only up to a normalizing constant, and $q$ (which, for us, represents the marginal distribution of a particle at the end of inference) is not known in closed form. To address the first difficulty, we instead measure the *approximation quality* $\log Z - \text{KL}(q \parallel g)$; because $Z$ is a function only of $g$ and not of the inference algorithm, as we vary the inference algorithm, the resulting variation in approximation quality is caused solely by changes to the KL term. To address the second difficulty, we work over extended state spaces to obtain lower bounds on the approximation quality (Lew et al., 2022; Zhao et al., 2024). In Appendix D.1 we show how to estimate this quantity in a general setting, and in Appendix D.2 we explain how to adapt this result to the setting where $g$ incorporates hard constraints (making the KL divergence discussed above potentially infinite).

### D.1 IMPORTANCE SAMPLING AND SEQUENTIAL MONTE CARLO

We begin by considering the case of a generic importance sampling algorithm, which draws $K$ particles $\boldsymbol{x}^{(i)} \in \mathcal{A}^*\text{EOS}$ from a proposal distribution $q$ to approximate the target distribution $\sigma$. For each particle, the algorithm computes an importance weight relative to the *unnormalized* target distribution $\tilde{\sigma}(\boldsymbol{x}) = Z_\sigma \sigma(\boldsymbol{x})$, where $Z_\sigma > 0$ is the unknown normalizing constant.

**Estimating the quality of 1-particle inference.** A very simple inference strategy is to use the proposal $q$ to generate a single sample $\boldsymbol{x}$, without further correction. A reasonable measure of inference quality would be $\text{KL}(q \parallel \sigma)$. However, due to the unknown normalizing constant of $\tilde{\sigma}$, the quantity that we can more easily estimate unbiasedly is $\log Z_\sigma - \text{KL}(q \parallel \sigma) = \mathbb{E}_{\boldsymbol{x} \sim q}[\log \tilde{\sigma}(\boldsymbol{x}) - \log q(\boldsymbol{x})]$. As the expression suggests, we can estimate this quantity by sampling many sequences $\boldsymbol{x}$ independently from $q$ and computing the average across samples of $\log \tilde{\sigma}(\boldsymbol{x}) - \log q(\boldsymbol{x})$. Because $Z_\sigma$ depends on only $\tilde{\sigma}$, and not the inference algorithm, we can compare proposals $q$ by estimating this quantity; higher is better, implying lower $\text{KL}(q \parallel \sigma)$.

**Estimating the quality of $K$-particle IS.** In $K$-particle IS, we consider the posterior approximation to be the distribution obtained by running importance sampling and resampling a particular particle $\boldsymbol{x}^{(k)}$ with probabilities proportional to the normalized particle weights. Formally, let $\mathbf{S} = \mathcal{A}^*\text{EOS} \times \mathcal{A}^*\text{EOS}^{K-1}$ be the space of $K$-particle collections with a distinguished, chosen particle; we write elements of $\mathbf{S}$ as $\langle \boldsymbol{x}^{(k)}, \boldsymbol{x}_{-k} \rangle$, where $\boldsymbol{x}_{-k} = \{\boldsymbol{x}^{(i)} \mid i = 1 \cdots K, i \neq k\}$ are the $K - 1$ unchosen particles. The importance sampling procedure defines the following distribution over $\mathbf{S}$:

$$q_{\text{IS}}(\langle \boldsymbol{x}^{(k)}, \boldsymbol{x}_{-k} \rangle) \overset{\text{def}}{=} \frac{w(\boldsymbol{x}^{(k)})}{\sum_{i=1}^{K} w(\boldsymbol{x}^{(i)})} \prod_{i=1}^{K} q(\boldsymbol{x}^{(i)}), \qquad (15)$$

where $w(\boldsymbol{x}^{(i)}) = \frac{\tilde{\sigma}(\boldsymbol{x}^{(i)})}{q(\boldsymbol{x}^{(i)})}$ is the importance weight for particle $\boldsymbol{x}^{(i)}$. We are interested in the quality of the *marginal posterior approximation*

$$q_{\text{IS}}^*(\boldsymbol{x}) \overset{\text{def}}{=} \sum_{\langle \boldsymbol{x}^{(k)}, \boldsymbol{x}_{-k} \rangle \in \mathbf{S}} q_{\text{IS}}(\langle \boldsymbol{x}^{(k)}, \boldsymbol{x}_{-k} \rangle) \cdot \mathbf{1}[\boldsymbol{x} = \boldsymbol{x}^{(k)}]. \qquad (16)$$

Although we can generate samples from $q_{\text{IS}}^*$ (by running importance sampling and selecting a chosen particle), we cannot evaluate its density exactly, so we cannot directly compute unbiased Monte

Carlo estimates of $\log Z_\sigma - \mathrm{KL}(q^*_{\mathrm{IS}} \parallel \sigma)$ as before. Instead, we extend the state space of the target distribution $\sigma$ to obtain a new distribution $\sigma_{\mathrm{IS}}$ over $\mathbf{S}$, following Andrieu & Roberts (2009):

$$\sigma_{\mathrm{IS}}(\langle \boldsymbol{x}^{(k)}, \boldsymbol{x}_{-k} \rangle) \stackrel{\text{def}}{=} \frac{1}{K}\sigma(\boldsymbol{x}_{-k} \mid \boldsymbol{x}^{(k)})\sigma(\boldsymbol{x}^{(k)}), \text{ where } \sigma(\boldsymbol{x}_{-k} \mid \boldsymbol{x}^{(k)}) \stackrel{\text{def}}{=} \prod_{i \neq k} q(\boldsymbol{x}^{(i)}) \quad (17)$$

We write $\tilde{\sigma}_{\mathrm{IS}} = Z_\sigma \tilde{\sigma}_{\mathrm{IS}}$ for the unnormalized extended target.

Note that the marginal distribution of the extended target,

$$\sigma^*_{\mathrm{IS}}(\boldsymbol{x}) = \sum_{\langle \boldsymbol{x}^{(k)}, \boldsymbol{x}_{-k} \rangle \in \mathbf{S}} \sigma_{\mathrm{IS}}(\langle \boldsymbol{x}^{(k)}, \boldsymbol{x}_{-k} \rangle)\mathbf{1}[\boldsymbol{x} = \boldsymbol{x}^{(k)}] \quad (18)$$

is equal to $\sigma(\boldsymbol{x})$. The KL divergence between two joint distributions is lower-bounded by the KL divergence between their marginals, so we have $\mathrm{KL}(q^*_{\mathrm{IS}} \parallel \sigma) \leq \mathrm{KL}(q_{\mathrm{IS}} \parallel \sigma_{\mathrm{IS}})$.

Both $q_{\mathrm{IS}}$ and $\tilde{\sigma}_{\mathrm{IS}}$ have tractable joint densities, so we can estimate $\log Z_\sigma - \mathrm{KL}(q_{\mathrm{IS}} \parallel \sigma_{\mathrm{IS}})$ by repeatedly generating $\langle \boldsymbol{x}^{(k)}, \boldsymbol{x}_{-k} \rangle \sim q_{\mathrm{IS}}$ and computing the log density ratio, which simplifies into a log average particle weight.

**Proposition 2** (Estimating $\log Z_\sigma - \mathrm{KL}(q^*_{\mathrm{IS}} \parallel \sigma)$). *Consider $K$ particles $\boldsymbol{x}^{(i)}$ generated from the proposal $q$ and let $w(\boldsymbol{x}^{(i)}) = \tilde{\sigma}(\boldsymbol{x}^{(i)})/q(\boldsymbol{x}^{(i)})$ be their importance weights. Then $\mathbb{E}\left[\log \frac{1}{K}\sum_{i=1}^K w(\boldsymbol{x}^{(i)})\right]$ is a lower bound on $\log Z_\sigma - \mathrm{KL}(q_{\mathrm{IS}} \parallel \sigma)$.*

*Proof.*

$$\log Z_\sigma - \mathrm{KL}(q^*_{\mathrm{IS}} \parallel \sigma) \geq \log Z_\sigma - \mathrm{KL}(q_{\mathrm{IS}} \parallel \sigma_{\mathrm{IS}}) \tag{19a}$$

$$= \mathbb{E}_{q_{\mathrm{IS}}}\left[\log \frac{\tilde{\sigma}_{\mathrm{IS}}(\langle \boldsymbol{x}^{(k)}, \boldsymbol{x}_{-k} \rangle)}{q_{\mathrm{IS}}(\langle \boldsymbol{x}^{(k)}, \boldsymbol{x}_{-k} \rangle)}\right] \tag{19b}$$

$$= \mathbb{E}_{q_{\mathrm{IS}}}\left[\log \frac{\frac{1}{K}\tilde{\sigma}(\boldsymbol{x}^{(k)})\prod_{i \neq k} q(\boldsymbol{x}^i)}{q(\boldsymbol{x}^{(k)})\frac{w(\boldsymbol{x}^{(k)})}{\sum_{i=1}^K w(\boldsymbol{x}^{(i)})}\prod_{i \neq k} q(\boldsymbol{x}^{(i)})}\right] \tag{19c}$$

$$= \mathbb{E}_{q_{\mathrm{IS}}}\left[\log \frac{\frac{1}{K}\tilde{\sigma}(\boldsymbol{x}^{(k)})\prod_{i \neq k} q(\boldsymbol{x}^i)}{q(\boldsymbol{x}^{(k)})\frac{\tilde{\sigma}(\boldsymbol{x}^{(k)})/q(\boldsymbol{x}^{(k)})}{\sum_{i=1}^K w(\boldsymbol{x}^{(i)})}\prod_{i \neq k} q(\boldsymbol{x}^{(i)})}\right] \tag{19d}$$

$$= \mathbb{E}_{q_{\mathrm{IS}}}\left[\log \frac{\frac{1}{K}}{\frac{1}{\sum_{i=1}^K w(\boldsymbol{x}^{(i)})}}\right] \tag{19e}$$

$$= \mathbb{E}_{q_{\mathrm{IS}}}\left[\log \frac{1}{K}\sum_{i=1}^K w(\boldsymbol{x}^{(i)})\right] \tag{19f}$$

$\square$

Hence, $\log\left(\frac{1}{K}\sum_{i=1}^K w(\boldsymbol{x}^{(i)})\right)$ can be seen as a single sample estimate of $\log Z_\sigma - \mathrm{KL}(q^*_{\mathrm{IS}} \parallel \sigma)$, with negative bias. The precise bias can be shown to be $-\mathrm{KL}(q_{\mathrm{IS}}(\langle \boldsymbol{x}^{(k)}, \boldsymbol{x}_{-k} \rangle \mid \boldsymbol{x}^{(k)}) \parallel \sigma_{\mathrm{IS}}(\langle \boldsymbol{x}^{(k)}, \boldsymbol{x}_{-k} \rangle \mid \boldsymbol{x}^{(k)}))$, which decreases as the number of particles increase (Lew et al., 2022). As before, the term $\log Z_\sigma$ is a constant that is independent of the inference algorithm, so this estimate can be used as a measure of inference quality across algorithms, where higher is better.

**Estimating the quality of SMC.** The same logic as above can be applied to SMC (Andrieu & Roberts, 2009; Lew et al., 2022; Zhao et al., 2024). As in the IS case, we use an extended target $\sigma_{\mathrm{SMC}}$ such that the density ratio works out to exactly the average particle weight at the end of the algorithm.

### D.2 ESTIMATING INFERENCE QUALITY FOR REJECTION-SAMPLED VARIANTS

When attempting to estimate the discrepancy between samples for algorithms $q_{\mathrm{alg}}$ and $g$, one difficulty is that $\mathrm{KL}(q_{\mathrm{alg}} \parallel g)$ can be infinite when $g$ incorporates hard constraints. This is because, in those

cases, there is positive probability on the outcome that all generated proposals fail to meet the constraints, and thus have mass 0 under $g$. A potential solution to this issue, explored by Zhao et al. (2024), is to instead estimate $\mathrm{KL}(g||q_{\mathrm{alg}})$. But this requires exact samples from $g$, which are impractical to obtain in our setting. We thus take a different approach and consider rejection-sampled versions of each of our algorithms, $q_{\mathrm{alg}}^{\mathrm{r}}$, which draw samples $\boldsymbol{x} \sim q_{\mathrm{alg}}^{\mathrm{r}}$ repeatedly until $q_{\mathrm{alg}} > 0$. In this case, we have

$$\mathrm{KL}(q_{\mathrm{alg}}^{r}||g) = \mathbb{E}_{q_{\mathrm{alg}}^{r}}\left[\log \frac{q_{\mathrm{alg}}^{\mathrm{r}}(\boldsymbol{x})}{g(\boldsymbol{x})}\right] \tag{20a}$$

$$= \mathbb{E}_{q_{\mathrm{alg}}^{r}}\left[\log \frac{q_{\mathrm{alg}}(\boldsymbol{x})}{g(\boldsymbol{x})Z_{alg}^{r}}\right] \tag{20b}$$

$$= \mathbb{E}_{q_{\mathrm{alg}}^{r}}\left[\log \frac{q_{\mathrm{alg}}(\boldsymbol{x})}{g(\boldsymbol{x})}\right] - \log Z_{alg}^{r} \tag{20c}$$

where $Z_{alg}^{r}$ is the acceptance rate of $q_{\mathrm{alg}}^{\mathrm{r}}$ (which we can estimate with standard Monte Carlo), and we can estimate the first term in Eq. (20c) up to an instance-specific constant by the derivations above.

| Table | Columns |
|---|---|
| singer | singer_id, name, … |
| concert | concert_id, concert_name, … |

(a) Example schema

| Query | $\Phi_{\text{eff}}$ | $\Phi_{\text{exp}}$ | Description |
|---|:---:|:---:|---|
| SELECT song_id FROM singer … | ✗ | ✗ | Invalid column name |
| SELECT singer_id FROM concert … | ✓ | ✗ | Invalid column name for table |
| SELECT singer_id FROM singer … | ✓ | ✓ | Valid column name for table |

(b) Example queries and potential values

Table 10: Overview of potentials used in Spider experiments for a given schema. We condition the base language model using two potentials. $\Phi_{\text{eff}}$ ensures syntactically valid SQL queries that only include table and column names present in the schema. $\Phi_{\text{exp}}$ further restricts SQL queries by ensuring a correct correspondence between column and table names.

# E    DOMAIN DETAILS

This section provides further details on the domains used in the experiments.

## E.1    TEXT-TO-SQL (SPIDER)

Spider is a large-scale text-to-SQL dataset of natural language questions and database schemas. Given a natural language question and schema, the task is to generate a valid SQL query that is semantically equivalent to a ground-truth query. In this domain, $\Phi_{\text{eff}}$ is used to enforce valid SQL syntax according to the SQL grammars released by Roy et al. (2024). These grammars include schema-specific constraints that limit table and column names to those present in the given schema but do not ensure correct table-column associations. Thus, we use $\Phi_{\text{exp}}$ to verify whether the (partial) SQL query references a column that exists in a table, returning 0 in the case that it does not and 1 otherwise. Table 10 provides an example of $\Phi_{\text{eff}}$ and $\Phi_{\text{exp}}$ applied to an SQL query. Since $\Phi_{\text{exp}}$ is only semantically meaningful when a generated query has fully specified the necessary table or alias information to check correspondences, we only run the table-column verification at clause boundaries once the FROM clause has been completed. We evaluate on the development split of Spider with execution accuracy, which checks whether the predicted SQL query's output matches that of the ground-truth query. We define $p(\boldsymbol{x})$ by prompting Llama-3.1 (8b) instruct with 3 examples followed by a rendering of the database and the natural language question.

## E.2    MOLECULAR SYNTHESIS (GBD-17)

A recent line of work has applied LMs to the problem of molecular synthesis, with the aim of generating candidate molecules with properties similar to molecules from known databases (see Oliveira et al., 2022, for review)—most commonly (e.g. Flam-Shepherd et al. (2022); Wang et al. (2024)) by prompting with examples of molecules in SMILES format (Weininger, 1988). We follow this approach, constructing prompts from random subsets of 20 molecules from the GDB-17 dataset (Ruddigkeit et al., 2012). We evaluate generations using the standard molecule fitness function Quantitative-Estimated Drug-likeness (QED; Bickerton et al., 2012) implemented in the Python RDKit library (Landrum, 2024). This metric combines eight physicochemical properties of a compound: Molecular weight, LogP, H-bond donors, H-bond acceptors, Charge, Aromaticity, Stereochemistry, and Solubility. Here, $\Phi_{\text{eff}}$ enforces SMILES syntax. To enforce properties not encoded by this syntax, we define $\Phi_{\text{exp}}$ using a molecule validator that can be applied to partial SMILES strings, implemented in the Python *partialsmiles* library (O'Boyle, 2024). The validator checks the SMILES prefix to ensure that the atom's valences are in a list of allowed valences, and attempts to find alternating patterns of single and double bonds to cover all aromatic systems in the partial string. The additional metrics reported in Fig. 4 are *Validity* (proportion of valid SMILES), *Weight* (exact molecular weight), *De Novo Similarity* (average pairwise Tanimoto similarity to the target distribution, excluding exact duplicates), and *Diversity* (inverse average pairwise Tanimoto similarity among compounds generated by a particular method).

### E.3 GOAL INFERENCE (PLANETARIUM)

Recent work has explored using LMs for planning with languages like the Planning Domain Definition Language (PDDL) (Ghallab et al., 1998), by either generating plans directly (Silver et al., 2022; Wong et al., 2023; Ying et al., 2023; Zhang et al., 2024; Zhi-Xuan et al., 2024), or generating descriptions of a task's initial and/or goal conditions, which classical planning algorithms can use to search for plans (Liu et al., 2023; Xie et al., 2023; Guan et al., 2023). In the spirit of the latter, we use the Blocksworld tasks from the Planetarium benchmark (Zuo et al., 2024), which provides natural language descriptions of a task's initial and goal conditions along with their ground-truth symbolic representations in the STRIPS subset of PDDL (Fikes & Nilsson, 1971). The original dataset is extremely challenging, requiring the LM to output a full STRIPS description of tasks with up to 100 objects—Zuo et al. (2024) report fewer than 2% of the outputs of Gemma 1.1 7B to be even parseable. We therefore simplify the task by limiting our evaluation to examples with fewer than 10 objects, and requiring the LM to generate only the goal conditions by appending the ground-truth initial conditions to the pre-prompt. Here, $\Phi_{\text{eff}}$ encodes STRIPS syntax for goals within Planetarium's Blocksworld domain definition; $\Phi_{\text{exp}}$ uses a gold-standard plan known to satisfy the ground-truth task description, and calls the VAL plan validator (Howey et al., 2004) to test whether partial goal descriptions are valid according to that plan. The gold-standard plans for each instance were derived using the fast downward algorithm (Helmert, 2006). Note that it is only possible to apply this $\Phi_{\text{exp}}$ potential to partial strings if goal descriptions are *monotonic*, that is, if any goal prefix describes a superset of the states that the full goal describes. This is the case for STRIPS, where goals must be described as conjunctions of literals, so that we can evaluate the potential after each literal in the conjunction is completed.

### E.4 DATA SCIENCE (DS1000)

DS-1000 (Lai et al., 2023) is a challenging code generation benchmark on data science problems in Python, split into subsets requiring the use of six popular libraries: Pandas, NumPy, Scikit-Learn, SciPy, TensorFlow, and PyTorch. For each problem instance, the language model is prompted with an English description of the problem and a sample test case in Python and is tasked with generating code that solves the problem and passes the test case. Each test case includes a result variable, and success depends on the execution. In preliminary experiments, we observed that our language model was able to generate syntactically correct Python programs for every sample. We, therefore, set $\Phi_{\text{eff}} = 1$ for our experiments in this domain. Thus, unlike the other three domains, the proposal distribution for all evaluations of DS1000 was simply $p$. In this domain, $\Phi_{\text{exp}}$ simply executes the test cases provided in the prompts from Lai et al. (2023) on generated (partial) Python programs, and returns 1 if no errors are produced and 0 otherwise (in particular, we did not make use of the output of test cases). Note that it is only possible to execute Python code when the generated sequence $x$ consists entirely of well-formed Python statements, thus in this domain $\Phi_{\text{eff}}$ can only be meaningfully applied at the boundary of statements—this motivates aligning SMC particles using statements as their steps, as explained in the "Further extensions" section in (§2).

# F    RELATED WORK

Our contributions are primarily situated among several bodies of work.

First, there is a large body of work leveraging LMs for semantic parsing or code generation tasks, while forcing adherence to a grammar or other constraints (Shin et al., 2021; Scholak et al., 2021; Poesia et al., 2022; Shin & Van Durme, 2022; Geng et al., 2023; Zheng et al., 2024; Moskal et al., 2024; Wang et al., 2024; Ugare et al., 2024). Closely related is a series of algorithmic advances that enable the efficient construction and application of grammar constraints for sequential inference problems via compilation to automata (Deutsch et al., 2019; Willard & Louf, 2023; Kuchnik et al., 2023; Koo et al., 2024). This space is further discussed in Appendix F.1.

Second, there is a large body of work that aims to generate from LMs subject to hard or soft constraints. Many such strategies are based in reinforcement learning (RL) (Ziegler et al., 2019; Stiennon et al., 2020; Bai et al., 2022; Ouyang et al., 2022), classifier-guided control (Cheng et al., 2024), efficient probabilistic inference through tractable proxy models (Zhang et al., 2023a), and locally applied *logit biasing* or *masking* based on domain-specific potential functions (Pascual et al., 2021; Huang et al., 2024). This is further discussed in Appendix F.2. In addition, there is a growing set of approaches that cast constrained sequential generation as probabilistic conditioning, leveraging the toolkit of probabilistic inference to derive principled generation strategies (Miao et al., 2020; Yang & Klein, 2021; Lew et al., 2023; Zhao et al., 2024; Park et al., 2025; Puri et al., 2025). These works themselves are motivated by a rich history of conditioning autoregressive models in NLP, often in combination with particle-based inference methods (Börschinger & Johnson, 2011; Dubbin & Blunsom, 2012; Yang & Eisenstein, 2013; Buys & Blunsom, 2015; Lin & Eisner, 2018). A discussion of the most relevant works is expanded in Appendix F.3.

In this work, our algorithms aim to unify and improve on each of these bodies of preceding work, tackling semantic parsing and code generation tasks with a combination of grammar constraints ($\Phi_{\text{eff}}$), expensive potentials ($\Phi_{\text{exp}}$), and asymptotically correct inference (via SMC).

## F.1    GRAMMAR-CONSTRAINED SEMANTIC PARSING WITH LMS

Shin et al. (2021) presented a system allowing LMs to be locally intersected with (boolean) CFGs to restrict generations to conform to target formal languages, and that with only a few in-context examples, such an inference-time strategy could outperform more substantial fine-tuning. Concurrently, PICARD (Scholak et al., 2021) presented an approach for intersecting LMs with an incremental parsing algorithm and showed how additional context-sensitive constraints could be imposed, such as requiring table-column matching for SQL generation via the use of programmable "guards". Synchromesh (Poesia et al., 2022) generalized these frameworks and extended the idea of incremental guards that can impose semantic restrictions during generation—such as typing and scoping rules—by dynamically constructing constraints as regular expressions on the fly. A great deal of other work has explored variants of LM-grammar intersection including the effectiveness of pre-training models on code for these settings (Shin & Van Durme, 2022), the runtime compilation of individual task instances into highly specific, task-specialized grammars (Geng et al., 2023), and even using the LM to generate grammars directly at runtime, that then restrict their own generation to solve a task (Wang et al., 2024). Other work has focused more closely on the standard syntactic-constraint problem but with an emphasis on optimizing efficient data structures and algorithms for fast LM-CFG intersection (Ugare et al., 2024; Zheng et al., 2024; Moskal et al., 2024).

A parallel line of work in this space has been concerned with the efficient construction and application of constraints for sequential inference problems. Deutsch et al. (2019) first noted that regular and context-free grammar constraints could be pre-compiled to automata—these could then be used during sequential inference to impose constraints with near-zero runtime overhead. This approach was independently developed and efficiently implemented in the context of restricting LM generations to regular expressions by the Outlines (Willard & Louf, 2023) and ReLM libraries (Kuchnik et al., 2023). Similar work was later developed by Koo et al. (2024), who extended several formal automata-theoretic characteristics of these constructions.

This work has noted the complications of efficiently intersecting grammars whose atoms are terminals and LMs whose atoms are tokens, which we refer to as the *token–terminal alignment problem*. An efficient and accurate solution to this problem space was one of several desiderata for our proposal

algorithm (see Alg. 1 in Appendix C for more details). These works have also discussed considerations that arise in the construction of automata, whose arcs are tokens, in the assignment of probabilities to strings. Namely, there are exponentially many latent token trajectories that correspond to a generated sequence. While the correct method for assigning string probabilities involves marginalizing over these trajectories (Cao & Rimell, 2021), in practice, simply using the *canonical tokenization* accounts for the overwhelming majority of the probability mass and can be justified (Chirkova et al., 2023; Kuchnik et al., 2023; Berglund et al., 2024; Vieira et al., 2024). In the present work, we do not enforce this assumption and allow all token trajectories.

## F.2 Conditional Generation Subject to Constraints

Language models pre-trained on a next-word objective reflect the distribution of their pre-training corpora, but often the inference-time needs of tasks necessitate that LMs modify this base distribution.

One approach to this class of problems is fine-tuning or reinforcement learning via some set of data that more closely mirrors the target task, such as via reinforcement learning from human feedback (RLHF) (Ziegler et al., 2019; Stiennon et al., 2020; Bai et al., 2022; Ouyang et al., 2022), but this method comes with challenges such as hyperparameter sensitivity and distributional collapse (Zheng et al., 2023; Zhu et al., 2023; Xiong et al., 2024). Some of these drawbacks can be mitigated by utilizing on-policy data (Tajwar et al., 2024) and imposing a KL penalty that penalizes shifting an LM too far from its prior distribution, casting optimization as a variational inference problem (Korbak et al., 2022; Amini et al., 2025).

Another inference-time approach to controlled generation for an LM is via direct modification to the LM's sampling distribution. This may be done via controlling intermediate layer activations with classifier guidance (Cheng et al., 2024), guiding autoregressive generation with a proxy probabilistic model for which estimation of the conditional density is tractable (Zhang et al., 2023a), or most commonly by directly intervening on the final logits before sampling to impose intersection with a potential function. Pascual et al. (2021) presented an early variant of such *logit-biasing* to encourage the presence of predefined guide words in generations.

This pattern is employed more broadly for hard constraints via *logit-masking*, setting the probability associated with particular tokens to zero, forcing the LM to sample from a subset of its distribution over sequences. This approach is used in most of the grammar-constrained semantic parsing work outlined in the previous section. Most recently, there have been attempts to restrict and re-weight generations not only via grammars but through additional expensive potentials such as grounded affordances in robotics settings (Ahn et al., 2022; Huang et al., 2024). However, in all of these works, constraints are imposed greedily, resulting in a local product of experts construction, and care is not taken to appropriately target the implied global product of experts. It should then come as no surprise that while standard approaches to grammar-constrained generation have been successful, they have been far from a silver bullet (Tam et al., 2024).

## F.3 Approximate Posterior Inference in Large Language Models via Sampling

This leads to a third line of work that formulates constrained generation from language models as posterior inference (Zhang et al., 2023a), and employs approximate inference to sample from the desired target distribution. This is in contrast to yet another line of work that views constrained decoding as an *optimization* problem, and tackles it via search (Meister et al., 2020; Lu et al., 2021; Zhang et al., 2023b) or continuous optimization (Dathathri et al., 2019; Kumar et al., 2021).

Several approximate inference algorithms have been explored for generating constrained samples from LMs, including rejection sampling (Poesia et al., 2022), as well as MCMC (Miao et al., 2019; Hie et al., 2022; Zhang et al., 2020; Qin et al., 2022; Kumar et al., 2022; Du et al., 2024). A weakness of MCMC-based approaches is that they do not fully exploit the autoregressive factorization of modern language models; each edit to a candidate sequence requires re-evaluating the entire sequence (or at least the entire suffix) to compute a new target density.

Lew et al. (2023) propose SMC steering of LMs via probabilistic programming specifications. This work enables provably accurate posterior sampling from such conditional targets, globally steering generation while only ever computing local constraints. Our approach builds on their work.

Shortly thereafter, Zhao et al. (2024) independently developed a framework for expressing various LM tasks as probabilistic inference problems that can be tackled with SMC. Similar to our work, Zhao et al. (2024) guide SMC with intermediate targets—in their case, learned twist functions via a novel contrastive method—that enable estimation of the expected future value of each candidate partial sequence. Their work also developed methods for evaluating LM inference algorithms via bi-directional bounds on the log-partition function that can be used to estimate the KL divergence between the inference and target distribution. In contrast to this prior work, our approach to SMC leverages incremental static and dynamic analyses to inform our proposal distributions and twist functions, as opposed to learning components of these algorithms via a costly contrastive fine-tuning procedure. In addition, our results directly relate the quality of our posterior approximation to improved performance on a series of standard, difficult benchmark tasks.

Concurrent with our work, Park et al. (2025) have highlighted the distinction between the prevalent locally constrained decoding approach and the more accurate targeting of the global distribution that arises from combining language models with constraints. Park et al. (2025)'s approach to approximate the global distribution is based on the concept of *expected future grammaticality*, which is the probability that the completion to be sampled from the LM will be compliant with the given grammar. The authors describe an iterative algorithm that approximates the global distribution by refining the estimates of the expected future grammatically. However, the proposed strategy shows relatively slow convergence, was specifically designed for a CFG constraint, and may not be easily adaptable to constraining with multiple potential functions.

