# OpenReview forum: "Syntactic and Semantic Control of Large Language Models via Sequential Monte Carlo"
_ICLR.cc/2025/Conference — ICLR 2025 Oral_

### Official Review · Reviewer_4hyJ · 2024-11-03

**Soundness:** 3
**Presentation:** 3
**Contribution:** 3
**Rating:** 8
**Confidence:** 3

**Summary:**

The paper introduces a Sequential Monte Carlo (SMC) approach for constrained generation in language models to address semantic and syntactic constraints that traditional methods fail to handle effectively. The novel elements include integrating domain-specific constraints via potential functions, weight correction to mitigate the bias from locally greedy decoding, and adaptive resampling that focuses computational effort on promising sequences. The method was tested on four challenging tasks: Python code generation, text-to-SQL translation, goal inference, and molecular synthesis. The experiments compared the SMC-based approach to several baselines and showed that incorporating weight correction and semantic potentials significantly improved performance, while adaptive resampling further enhanced results by reallocating resources to better particles. The total SMC method outperformed the ablated SMC variants on the selected task.

**Strengths:**

-Good motivation from analysis of weight formulations for importance sampling.

-Benchmarks validate claims that the proposed algorithmic components improve downstream performance. Additionally, authors chose a sensible set of benchmarks.

-Weight correction and resampling seem to be novel components.

**Weaknesses:**

-The method was not benchmarked against alternative methods. While the ablation study is useful, how does the method compare against other SMC-based techniques such as the ones cited in the related works section that are particularly relevant to this work? E.g. comparisons against the method in Lew et al. and Zhao et al. would be beneficial. There are other non-SMC-based methods that could also be benchmarked against.

-Only Llama 3.1 8-B was evaluated. The manuscript would benefit from benchmarks on additional LLMs to see if results are consistent across similar sized LLMs. I would be curious to see if the benefits are as substantial on larger models, but I understand the authors may have limited computational resources for such analyses.

-There is a lack of theoretical grounding as to the benefits of the components. E.g., a theorem rigorously showing the reduction in KL-Divergence shown in Figure 2 would strengthen the manuscript.

-Notation can be difficult to follow at times. Exposition can be a bit drawn out in certain places, e.g. section 2. I appreciate the authors trying to point out the inefficiencies in each component of MC in order to justify their approach, but I think the exposition would benefit from a condensed explanation of, e.g., the computational burdens of IS.

**Questions:**

-Is there a difference between the right part of Figure 1 and Table 2? I would suggest the authors keep only one to avoid redundancy.

-Why was the instruct version of Llama 3.1 used for the SQL task but not the others?

---

> ### Author Response · Authors · 2024-11-22
> **Response to 4hyJ**
>
> > The method was not benchmarked against alternative methods. While the ablation study is useful, how does the method compare against other SMC-based techniques such as the ones cited in the related works section that are particularly relevant to this work? E.g. comparisons against the method in Lew et al. and Zhao et al. would be beneficial. There are other non-SMC-based methods that could also be benchmarked against.
>
> - See [C. Response to questions about comparisons to other methods](https://openreview.net/forum?id=xoXn62FzD0&noteId=ercx7hhxwz)
> - See [A. Response to questions regarding number of particles](https://openreview.net/forum?id=xoXn62FzD0&noteId=TCZB8KqXvx)
> - See [D. Response to questions about relationship with Lew et al (2023) and Zhao et al. (2024)](https://openreview.net/forum?id=xoXn62FzD0&noteId=ercx7hhxwz)
>
> > Only Llama 3.1 8-B was evaluated. The manuscript would benefit from benchmarks on additional LLMs to see if results are consistent across similar sized LLMs. I would be curious to see if the benefits are as substantial on larger models, but I understand the authors may have limited computational resources for such analyses.
>
> See [B. Response to questions about evaluating different LMs](https://openreview.net/forum?id=xoXn62FzD0&noteId=9NlTwE2DfC)
>
> > There is a lack of theoretical grounding as to the benefits of the components. E.g., a theorem rigorously showing the reduction in KL-Divergence shown in Figure 2 would strengthen the manuscript.
>
> In general, the question of how the required sample size for SMC scales with the KL divergences between intermediate proposals and intermediate targets is a difficult, high-value open question in the field: even the analogous result for the much simpler setting of Importance Sampling was a significant landmark, only established recently \[Chatterjee and Diaconis, 2018\]. While we are very interested in this question, we opted not to explore it deeply since it falls outside the scope of the main goals of the paper. However, we note that Appendix D includes derivations of the KL bounds between our different methods and the global product target posterior.
>
> > Notation can be difficult to follow at times. Exposition can be a bit drawn out in certain places, e.g. section 2\. I appreciate the authors trying to point out the inefficiencies in each component of MC in order to justify their approach, but I think the exposition would benefit from a condensed explanation of, e.g., the computational burdens of IS.
>
> We agree with the reviewer on the unevenness of the exposition in section 2. We took their suggestion and significantly condensed the explanation of the computational burdens of IS — see lines 180-186. We also made additional edits to tighten the prose in other parts of sections 2 and 3 — see for instance the introduction to SMC in lines 188-94,  and the introduction to section 3\.
>
> > Is there a difference between the right part of Figure 1 and Table 2? I would suggest the authors keep only one to avoid redundancy.
>
> The right part of Figure corresponds to the methods which are the cumulative addition of the algorithmic components of our approach (grammar constraint, weight corrections, semantic potentials, and resampling). Table 2 also reports accuracy for these methods, but includes additional ablations, such as “LM w/ grammar constraint, potential (Sample-rerank)” and “LM w/ grammar constraint, correction, and resampling (Grammar SMC)”.
>
> > Why was the instruct version of Llama 3.1 used for the SQL task but not the others?
>
> We used the instruct version of Llama 3.1 because the Spider dataset makes use of a chat template for prompt formatting, which is better suited for instruct-tuned LMs. This variation had the added benefit of allowing us to show our method’s performance under both base and instruction tuned models. In the other domains, we opted for the base models as reinforcement-learning from human feedback has been shown to be detrimental to calibration \[1, 2\]
>
> \[1\] Kadavath, Saurav, et al. "Language models (mostly) know what they know." *arXiv preprint arXiv:2207.05221* (2022).
>
> \[2\] Achiam, Josh, et al. "Gpt-4 technical report." *arXiv preprint arXiv:2303.08774* (2023).

---

> > ### Comment · Reviewer_4hyJ · 2024-12-03
> >
> > > - See C. Response to questions about comparisons to other methods
> >
> > Okay, great! This makes the comparisons to other methods more clear.
> >
> > > - See D. Response to questions about relationship with Lew et al (2023) and Zhao et al. (2024)
> >
> > Thank you for the clarification here. Question about Table 6: why are the particle sizes different here? Doesn't Table 5 have the 5 particle results for Text-to-SQL? The results only look marginally better given the confidence intervals.
> >
> > > - See B. Response to questions about evaluating different LMs
> >
> > This shows the effectiveness with scale, albeit on a single task. Thank you for trying out the larger model.
> >
> > > In general, the question of how the required sample size for SMC scales with the KL divergences between intermediate proposals and intermediate targets is a difficult, high-value open question in the field: even the analogous result for the much simpler setting of Importance Sampling was a significant landmark, only established recently [Chatterjee and Diaconis, 2018]. While we are very interested in this question, we opted not to explore it deeply since it falls outside the scope of the main goals of the paper. However, we note that Appendix D includes derivations of the KL bounds between our different methods and the global product target posterior.
> >
> > Thank you for elaborating on the theoretical work. At the very least, the KL derivation is useful to have in writing here.
> >
> > The remaining concerns about the manuscript's readability have been addressed. Overall, I think the paper would be a good contribution to this venue. I have adjusted my score accordingly.

---

> > > ### Author Response · Authors · 2024-12-03
> > >
> > > Thank you for your positive feedback and response! To address your question:
> > >
> > > > Question about Table 6: why are the particle sizes different here? Doesn't Table 5 have the 5 particle results for Text-to-SQL? The results only look marginally better given the confidence intervals.
> > >
> > > Table 6 reports a comparison between our method (using multinomial resampling) with 10 particles and SMC steering with 5 particles and a beam size of 3. This configuration gives SMC steering an effective particle count of N = 15, which actually provides it an advantage in the comparison to our method with N = 10 particles. Nonetheless, we agree that even in the case of our method with 10 particles, the results are only slightly better.
> > >
> > > However, we emphasize that our approach is agnostic to the particular resampling method that is used. The SMC steering method is completely compatible with our framework---we can swap out multinomial resampling with their without-replacement resampling. There are also other resampling methods that have been explored in the literature that can be integrated with our framework. We opted to use multinomial resampling in our experiments because it is currently the most standard approach, and did not intend to advocate for one resampling method over the other. We will aim to make this clearer in any subsequent versions of the paper.

---

### Official Review · Reviewer_8wBr · 2024-11-04

**Soundness:** 3
**Presentation:** 4
**Contribution:** 3
**Rating:** 8
**Confidence:** 4

**Summary:**

The paper concerns inference-time approaches for controlled generation with language models. Building on [Lew et al 2023], the authors develop a method based on sequential Monte-Carlo. The method proceeds segment-by-segment, first extending the segment (e.g. generating a next-token) subject to a token-level score (e.g., arising from a grammar constraint), then reweighting the candidates based on a partial sequence score (e.g., whether the code-so-far has a runtime error), then determining the next set of candidates by resampling candidates based on their weights.

The authors evaluate the method using prompted Llama 3 (base or instruct, varied by task) on 4 tasks. In each task, they construct task-specific token-level and partial-sequence level potentials, and provide an ablation of each of their three proposed components. The authors study the divergence between the target distribution and the distribution induced by running the algorithm, finding that each component leads to a lower KL divergence. Additionally, they provide a nice visualization of the sampling distributions and target distributions for their molecular generation task, and show that samples from their method improve along various dimensions.

In general the paper is written quite precisely, and several intuitive ideas (e.g., token masking or filtering out a sequence if its code doesn't run) are placed into a probabilistic framework.

**Strengths:**

Originality
- To my knowledge, the extension of sequential Monte-Carlo to the task settings in the paper, and the specific generation receipe (re-weighting, resampling) are new. However, I am not closely familiar with [Lew et al 2023] or its subsequent papers (which are mentioned several times by the authors). Therefore, my evaluation of novelty may be slightly off.
- Placing ideas such as token-masking, filtering out partial sequences, and selecting partial sequences to explore next in a probabilistic framework is a nice contribution (with the same caveats in the point above).

Quality
- The experimental evaluation presents a controlled study that ablates each component in the model.
- Derivations and the divergence analysis seem to be of high quality.

Clarity
- Once the reader becomes familiar with the terminology, the paper is written clearly and precisely.

Significance
- The method could potentially be useful in settings where token-level and partial-sequence level constraint functions are available (e.g., those in the experiments). This has some generality (though could also be viewed as a limiting factor).
- Placing more domains and settings into the probabilistic framing from [Lew et al 2023] helps to further the probabilistic perspective on sequence generation.

**Weaknesses:**

My primary concerns were on the experimental validation. The paper performs a self-contained, controlled experiment using one Llama 3 model on a set of tasks. As a result, it was unclear how the findings generalize to other models, or how they compare in terms of performance to other methods in the literature.

1) For example, taking the example of DS-1000, the absolute numbers are quite low: the DS-1000 paper reports up to 39.2 performance (with Codex-002) versus 28.5% here (with Llama 3). These are *not* comparable since they use different models, but it would be nice to see how this method performs for models closer to the state of the art. Similarly, Lever [1] reports numbers on Spider from previous work that range from 67.0% to 81.9% [1]. The reason this is important is that the exact experiment setup can lead to different conclusions on the performance of methods, so it was concerning that the absolute numbers seemed low. However, the authors could potentially clarify this.

2) It was also unclear why 10 particles was selected, since in these sampling methods the number of samples can impact performance, and we often want to understand how performance varies with the sampling budget. How does the method vary as the number of particles varies? Is there a sample-and-rerank approach that could outperform this method if it drew a large number of samples?

[1] LEVER: Learning to Verify Language-to-Code Generation with Execution, Ni et al ICML 2023

**Questions:**

Please see the questions in the Weaknesses above.

---

> ### Author Response · Authors · 2024-11-22
> **Response to reviewer 8wBr**
>
> > However, I am not closely familiar with \[Lew et al 2023\] or its subsequent papers (which are mentioned several times by the authors). Therefore, my evaluation of novelty may be slightly off.
>
> See [D. Response to questions about relationship with Lew et al (2023) and Zhao et al. (2024)](https://openreview.net/forum?id=xoXn62FzD0&noteId=ercx7hhxwz)
>
> > My primary concerns were on the experimental validation. The paper performs a self-contained, controlled experiment using one Llama 3 model on a set of tasks. As a result, it was unclear how the findings generalize to other models, or how they compare in terms of performance to other methods in the literature.
>
> - See [B. Response to questions about evaluating different LMs](https://openreview.net/forum?id=xoXn62FzD0&noteId=9NlTwE2DfC)
> - See [C. Response to questions about comparisons to other methods](https://openreview.net/forum?id=xoXn62FzD0&noteId=ercx7hhxwz)
>
> > It was also unclear why 10 particles was selected, since in these sampling methods the number of samples can impact performance, and we often want to understand how performance varies with the sampling budget. How does the method vary as the number of particles varies?
>
> See [A. Response to questions regarding number of particles](https://openreview.net/forum?id=xoXn62FzD0&noteId=TCZB8KqXvx)
>
> > For example, taking the example of DS-1000, the absolute numbers are quite low: the DS-1000 paper reports up to 39.2 performance (with Codex-002) versus 28.5% here (with Llama 3). These are *not* comparable since they use different models, but it would be nice to see how this method performs for models closer to the state of the art. Similarly, Lever \[1\] reports numbers on Spider from previous work that range from 67.0% to 81.9% \[1\]. The reason this is important is that the exact experiment setup can lead to different conclusions on the performance of methods, so it was concerning that the absolute numbers seemed low. However, the authors could potentially clarify this.
>
> - To address this point, we ran new experiments on DS-1000 using LLama3 70b, a model almost 9 times larger than the one used in our original experiments, and that has been shown to outperform GPT-3.5 turbo \[1\]. Even though this is still an open source model, and not as capable as the larger, closed-source, finetuned GPT-3 Codex model used in the DS-1000 paper (with the LLama3 70b base LM achieving 24% performance on DS-1000 vs Codex’s 39.2%), we find that our approach allows it to beat Codex, achieving 40.7% performance.
>
> > Is there a sample-and-rerank approach that could outperform this method if it drew a large number of samples?
>
> - In our experiment evaluating model performance as a function of sample size, we find sample and rerank approaches (named “LM w/ grammar constraint, potential) even with 5x more particles (in Goal Inference) and 10x more particles (in Data Science, Text-to-SQL and Molecular Synthesis) do not outperform our full SMC method.
> - We additionally note that the only condition under which SMC can perform worse than importance sampling (i.e. sample-and-rerank) is if resampling becomes detrimental. This can happen if the intermediate signals provided by the potentials are actively harmful when trying to steer the LM towards the correct distribution. This can only be the case if a user has designed their potentials poorly, since the fundamental goal of a potential is to provide a helpful intermediate signal. When using boolean potentials, this will never be the case, since the intermediate signal has no effect on resampling when $\phi(x) = 1$, only when $\phi(x) = 0$, and the definition of a potential dictates that $\phi(x) = 0 \implies \phi(xy) = 0$, meaning that they can never provide misleading signals.
>
> \[1\] Dubey, Abhimanyu, et al. "The llama 3 herd of models." *arXiv preprint arXiv:2407.21783* (2024).

---

> > ### Comment · Reviewer_8wBr · 2024-11-26
> >
> > Thank you for the detailed responses, my concerns have been addressed! I have raised my score to 8, and think that this paper should be accepted.

---

### Official Review · Reviewer_gr6N · 2024-11-08

**Soundness:** 3
**Presentation:** 3
**Contribution:** 3
**Rating:** 8
**Confidence:** 3

**Summary:**

The authors apply sequential Monte Carlo to tackle semantic parsing problems involving global constraints. They introduce an enhanced SMC approach, incorporating efficient stochastic approximations of full token-masking distributions and semantic potential to boost performance across five diverse datasets. Additionally, they estimate the KL divergence between each method’s output distribution and the global product-of-experts distribution, demonstrating that the latter is well-calibrated.

**Strengths:**

The authors propose adapting SMC methods to novel semantic parsing tasks, resulting in notable performance improvements.

The author conduct a interesting analysis and shows that resampling improves the approximation of the global product-of-experts distribution and approximation quality are consistent with those observed in downstream accuracy evaluation.

**Weaknesses:**

In terms of experiments:
- The authors do not emphasize their unique algorithmic contributions within the experiments. The authors could also report the performance of LM with grammar constraint, weight correction and resampling as a regular SMC baseline to further show the effectiveness of semantic potential. Additionally, the authors lack a detailed comparison between their method and the highly relevant SMC method in https://arxiv.org/pdf/2306.03081, and should report it as a baseline, e.g., including without-replacement resampling.
- For ablation studies, how the number of particles will affect the final performance should be analyzed.

There is no mention of the computation cost, it would be very useful if the authors could evaluate the efficiency of the proposed algorithm.

**Questions:**

See weaknesses.

Also minor writing issues for clarity, the authors should specify what is meant by "some" in lines 41-44 and clarify the specific applications referenced in lines 157-158.

---

> ### Author Response · Authors · 2024-11-22
> **Response to reviewer gr6N**
>
> > The authors do not emphasize their unique algorithmic contributions within the experiments.
>
> Thank you for pointing this out. We edited the introduction to section 3 to better highlight our unique algorithmic contributions, and highlighted how our ablations relate to previous methods in the literature, see \[Response to comparison to other methods\]
>
> > The authors could also report the performance of LM with grammar constraint, weight correction and resampling as a regular SMC baseline to further show the effectiveness of semantic potential.
>
> Thank you for this suggestion: we added this baseline to our experiments — see the updates to table 1 in the main paper, and tables 5 and 6 in the appendix, and point 3 in  \[Response to comparison to other methods\]
>
> > Additionally, the authors lack a detailed comparison between their method and the highly relevant SMC method in [https://arxiv.org/pdf/2306.03081](https://arxiv.org/pdf/2306.03081), and should report it as a baseline, e.g., including without-replacement resampling.
>
> See [D. Response to questions about relationship with Lew et al (2023) and Zhao et al. (2024)](https://openreview.net/forum?id=xoXn62FzD0&noteId=ercx7hhxwz)
>
> > For ablation studies, how the number of particles will affect the final performance should be analyzed.
>
> See [A. Response to questions regarding number of particles](https://openreview.net/forum?id=xoXn62FzD0&noteId=TCZB8KqXvx)
>
> > There is no mention of the computation cost, it would be very useful if the authors could evaluate the efficiency of the proposed algorithm.
>
> We added a discussion and small experiment on the computational cost of our approach to Appendix A.4 (line 899). We note that at every token, our SMC approach incurs two computational overheads relative to a simple locally-constrained decoding baseline: resampling, and computing semantic potentials. Though the cost of resampling is negligible, computing semantic potentials presents a more significant cost that varies across domains: our new experiment shows that that cost rarely rises above \~30ms per token.  In general, however, this cost is reduced by two factors: 1\) semantic potentials often need to run expensive computations not at every token, but only at larger, semantically meaningful units (for instance the end of a SQL clause or a python statement)---therefore significantly lessening the average cost per token, 2\) semantic potentials are often CPU rather than GPU computations (and so the cost of computation is much cheaper).
>
> > Also minor writing issues for clarity, the authors should specify what is meant by "some" in lines 41-44 and clarify the specific applications referenced in lines 157-158.
>
> Thank you for pointing these out. We have given examples of the signals mentioned by “some” in lines 41-44, and clarified the applications referenced in lines 157-158 by referencing Table 1, which lists the tasks and constraints which cannot be cheaply computed, and provided an example.

---

> > ### Comment · Reviewer_gr6N · 2024-11-25
> >
> > Thank you for your detailed responses, which address my previous questions and concerns. I increase my score accordingly.

---

### Official Review · Reviewer_MMNX · 2024-11-11

**Soundness:** 3
**Presentation:** 3
**Contribution:** 3
**Rating:** 8
**Confidence:** 5

**Summary:**

Controlling LLM generation to follow logical, syntactical or semantical (soft) constraints is a challenging task. The authors propose to unify controllable generation as sampling from the un-normalized distribution p_{LM}(x) \phi(x) where \phi is an energy function specifying the constraints. The authors propose to leverage sequential Monte Carlo to sample from the desired un-normalized conditional distribution. The authors conducted extensive evaluation of their approach on various downstream tasks with different combinations of constraints and  have demonstrated significant improvement compared to the LLM baseline. One important ablation study has suggested the positive correlation between approximation accuracy and generation quality, motivating for further research on improving the sampling algorithm or the proposal distribution.

**Strengths:**

This work solves one important problem with some of the widely adopted constrained/structured generation such as Guidance, SGLang and Outlines: that is, these framework achieves control by masking out next-tokens that would violate the constraint, leading to biased sampling (compared to the ground-truth conditional distribution). By leveraging sequential Monte Carlo, the proposed technique is able to approximate unbiased sampling in a relatively practical/scalable way. Empirical evaluations demonstrate strong performance on challenging real-world problems.

**Weaknesses:**

Some detailed analysis/case study on the sample complexity of SMC would provide more insights, especially how much better SMC is compared to naive importance sampling.

**Questions:**

What are the other (potentially better) proposal distributions? E.g. LLM fine-tuned/prompted with task-specific supervision.

---

> ### Author Response · Authors · 2024-11-22
> **Response to reviewer MMNX**
>
> Thank you for the time and effort spent on reviewing our work.  We appreciate your positive comments and assessment\!  We address your concerns and questions below.
>
> > Some detailed analysis/case study on the sample complexity of SMC would provide more insights, especially how much better SMC is compared to naive importance sampling.
>
> See [A. Response to questions regarding number of particles](https://openreview.net/forum?id=xoXn62FzD0&noteId=TCZB8KqXvx)
>
> > What are the other (potentially better) proposal distributions? E.g. LLM fine-tuned/prompted with task-specific supervision.
>
> We thank the reviewer for the interesting question. Indeed it is often possible to craft a better proposal distribution through offline computation (such as finetuning or prompt tuning), and trade it off against online computation (such as increasing the number of particles). We make two observations:
>
> * We focus on online computation in this paper because it is a more flexible setting: it works even if you don’t have a lot of training data, and does not require retraining if your objective changes.
> * Though finetuning and prompt tuning can lead to better sample efficiency, they can sometimes warp the LM distribution in ways that are not desirable (e.g. deteriorating calibration, see \[1, 2\] for discussions in the context of reinforcement learning from human feedback). An interesting direction enabled by our approach is to use different LMs for proposal and target, allowing sample efficiency without distorting the posterior.
>
> We also point out our new related experiment using both larger and smaller LMs, which is a way of varying the proposal distribution (and also the target) by making it better or worse. We find that our model can drastically improve the performance of smaller LMs, allowing them to outperform locally constrained decoding for LMs about 8 times their size, and allowing LLama 3 70B to outperform a larger, closed-source, fine-tuned model (GPT-3 Codex 002\): this suggests that our method for online computation can often perform as well as or better than offline computation (which is consistent with previous findings, see e.g. the best-of-K vs finetuning comparison in \[3\]).
>
> \[1\] Kadavath, Saurav, et al. “Language models (mostly) know what they know.” arXiv preprint arXiv:2207.05221 (2022).
> \[2\] Achiam, Josh, et al. “Gpt-4 technical report.” arXiv preprint arXiv:2303.08774 (2023).
> \[3\] Rafailov, Rafael, et al. “Direct preference optimization: Your language model is secretly a reward model.” Advances in Neural Information Processing Systems 36 (2024).

---

### Author Response · Authors · 2024-11-22
**New main takeaways**

We thank reviewers for their insightful comments: in response to common points brought up, we ran a new suite of experiments varying the base LM and the number of particles  — effectively quadrupling the number of LM experiments in the paper. We have edited the manuscript, highlighting changes in blue: the comprehensive new experiments are in Appendix A—we plan to incorporate them into the main text, but wanted to share them soon for discussion. We underscore key takeaways below:

- Our method allows smaller LMs to outperform LMs that are over 8 times larger, and allows an open source model (LLama 3 70B) to outperform a larger, closed-source, fine-tuned model (GPT-3 Codex 002)
- Our SMC method scales better with sample size than Importance Sampling, outperforming the latter even with 5 times (Goal Inference) or 10 times (text-to-SQL, Molecular Synthesis, Data Science) fewer particles.

_We post individual general responses in separate comments. Reviewer specific responses are as replies to specific reviews._

---

### Author Response · Authors · 2024-11-22
**A. Response to questions regarding number of particles**

We ran an extensive study of how the number of particles affect performance. As mentioned in our [New main takeaways](https://openreview.net/forum?id=xoXn62FzD0&noteId=qYuFwRrFsh), our SMC method scales better with sample size than Importance Sampling, outperforming the latter even with 5 times (Goal Inference) or 10 times (text-to-SQL, Molecular Synthesis, Data Science) fewer particles.
In general, we find that variations in performance across the number of particles depends on the domain. In both the Molecular Synthesis and the text-to-SQL domain, we do not find significant differences in performance across 5, 10 and 50 particles. For the Goal Inference and Data Science domains, on the other hand, we find that increasing the number of particles in general greatly improves performance across different ablations but especially for SMC. Interestingly, the performance gain from larger sample sizes occurs in the most challenging domains, whose semantic potentials run the complex procedures of executing a pddl plan or a python program.

---

### Author Response · Authors · 2024-11-22
**B. Response to questions about evaluating different LMs**

Additional models: In addition to our original experiments on a model with 8 billion parameters (Llama 3.1), we  ran additional experiments using both a larger 70 billion parameter LM (Llama 3) and a smaller 1 billion parameter LM (LLama 3.2). Due to computational resource constraints, we could only run the 70B experiment for one domain: we chose to do it for our most challenging domain, DS-1000. We find that:
- As expected, performance across the board is higher for larger LMs.
- The overall ranking of different approaches is preserved across the 1B, 8B, and 70B models: weight correction, resampling, and semantic potentials still generally improve performance, with the highest performing approach still being the combination of all three, that is SMC.
- Most interestingly, we find that, in general, the relative gains in accuracy provided by our method are more pronounced for smaller models: in fact, we find that in three out of our 4 domains (with the exception of text-to-SQL), our approach using the smaller model outperforms local decoding using a model 8 times larger, suggesting that our approach can dramatically improve the performance of smaller LMs. We also find, in DS-1000, that our approach allows an open source model (LLama 3 70B) to outperform a larger, closed-source, fine-tuned model (GPT-3 Codex 002, as reported by [1])

[1] Lai, Yuhang, et al. "DS-1000: A natural and reliable benchmark for data science code generation." International Conference on Machine Learning. PMLR, 2023.

---

### Author Response · Authors · 2024-11-22
**C. Response to questions about comparisons to other methods**

Comparison to other methods: We agree with the reviewers that the previous version of the paper did not make clear how our ablation studies related to other methods in the literature. To address this, we added a new baseline method (see table 1), and rewrote the introduction to section 3  and created new shorthands for the models to make the link more explicit. We summarize the above here:
1. LM + grammar constraints [Locally-constrained decoding] corresponds to the most common approach to grammar-constrained code generation with LMs (Shin et al., 2021; Scholak et al., 2021; Poesia et al., 2022; Willard & Louf, 2023; Moskal et al., 2024; Ugare et al., 2024). This approach applies the grammar constraint locally via token masking or token biasing at each step of generation.
2. LM + grammar constraints + semantic potentials [Sample-rerank] corresponds to a common approach for incorporating an external signal into an LM’s generations post-hoc, for instance a reward model (e.g. [1]), or a boolean encoding validity (e.g. [2]). This approach samples some number of full sequences from the locally-constrained model, then weights these using the semantic potential at the end.
3. LM + grammar constraints + weight correction + resampling: We add this new baseline, which we refer to as “Grammar-only SMC” as a shorthand, and note that it corresponds to a straightforward application of Lew et al. 2023 to incrementally correct locally-constrained decoding to the true Bayesian posterior via resampling; it is related to the method in Park et al. 2024, which also attempts to correct for the greediness of locally-constrained decoding. We find that, in general, the improvement of SMC without semantic potentials over Importance Sampling without semantic potentials is modest; the improvement is large only in the Goal Inference domain, where locally-constrained decoding seems to suffer the most from myopic sampling — however adding semantic potentials leads to large gains even here and still outperforms other methods, consistent with our previous findings.

[1] Nakano, Reiichiro, et al. "Webgpt: Browser-assisted question-answering with human feedback." arXiv preprint arXiv:2112.09332 (2021).

[2] Olausson, Theo X., et al. "LINC: A neurosymbolic approach for logical reasoning by combining language models with first-order logic provers." arXiv preprint arXiv:2310.15164 (2023).

---

### Author Response · Authors · 2024-11-22
**D. Response to questions about relationship with Lew et al (2023) and Zhao et al. (2024)**

Our work is complementary to both Lew et al. 2023 and Zhao et al. 2024.

Lew et al. 2023 is a workshop paper that proposes, but does not systematically evaluate, the use of SMC to perform inference in a very broad class of probabilistic sequence models. We will revise our paper to better contextualize our work with respect to theirs; here, we summarize several key points:

- One contribution of Lew et al. 2023 is a without-replacement resampling algorithm that they speculate may be useful in the language modeling setting, making SMC more similar to beam search without compromising its unbiasedness. This resampling algorithm is completely compatible with our contributions: we can replace multinomial resampling in our SMC implementation with their without-replacement resampling. We conducted preliminary experiments testing this in the Text-to-SQL domain, and found that without-replacement resampling slightly hurt performance. See Appendix, Table 6.

- Lew et al. 2023 do not give a general recipe for defining intermediate target distributions or proposals; instead, they describe three example instantiations of their SMC framework (for infilling, prompt intersection, and natural language generation subject to word-length constraints). Our work specializes and extends sequential Monte Carlo for code generation under syntactic and semantic constraints, with concrete recipes for defining proposals and intermediate targets in terms of these constraints. Note that our proposal distributions and weight calculations are not quite a special case of Lew et al. 2023’s formalization, because our incremental weights are not deterministic functions of the generated string (see Appendix B). Note also that although Lew et al. 2023 use the word “potential” in their technical exposition, it means something different for them; their “potential” is our “incremental weight.” They have no direct analogue of the potentials from our work.


- Although all examples in Lew et al. 2023 deal with natural language generation and not code generation, the “word-length constraints” example can be straightforwardly adapted to use grammar constraints, given an efficient incremental parser. The resulting algorithm could be seen as an ablation of our method that removes the semantic potentials, but keeps the weight correction and the resampling steps  [though it does not use our character-level proposal optimization in Appendix B]. This particular ablation was missing in our initial submission but we have now added it; see the portion of this response labeled “Comparison to other methods.”


- Finally, one contribution of Lew et al. 2023 was to propose language model probabilistic programs as a medium for specifying inference problems (as well as proposals and twist functions); given such a program, SMC can be automated. Conceptually, our method can be understood as a way to compile a set of user-provided potentials into such a probabilistic program. In order to practically implement our approach this way, the Python library of Lew et al. 2023 would have to be extended to support certain forms of CPU parallelism (which we use to efficiently run our incremental parser on multiple particles in parallel) and unbiased importance weight estimation (which we use to compute faster proposal distributions, see Appendix B).

Zhao et al. 2024 presents and thoroughly compares various methods for learning twist functions for natural-language generation tasks such as infilling. (To translate to our setting: if users define a set of potentials on full/complete strings, Zhao et al. 2024’s method can be understood as learning an extension of the potentials to cover partial strings.) Zhao et al. 2024 use these learned twist functions the same way that we use our efficient potentials $\phi_{eff}$: to inform both the proposal distributions and the intermediate targets in SMC.

However, learning twist functions can be very expensive. For each new constrained generation problem, twist functions need to be re-learned, a process that can take hours or days even for small models. (Note that twist learning is more like reinforcement learning than supervised fine-tuning, and may require many cycles of sampling roll-outs of the current SMC algorithm, then optimizing twists.) For this reason, all of Zhao et al. 2024’s experiments are run on either the 33M-parameter TinyStories model or, in one case, GPT-2 Medium (355M parameters). By contrast, our method requires no additional training, and was cheap enough that we could evaluate our method using 1B, 8B, and 70B-parameter variants of the Llama models. Indeed, a key insight of our approach is that although they may not be optimal twists, in structured generation domains we often have access to off-the-shelf tools that can serve as reasonable programmable twists, ranging from prefix parsers (for grammatical constraints) to more complex partial evaluators that incrementally capture more of the generated code’s semantics.

---

### Author Response · Authors · 2024-12-04
**Summary of rebuttal and discussion period**

We thank the reviewers for their insightful comments. There was a great deal of agreement between reviewers on the points that most needed improvement: we addressed their feedback and effectively quadrupled the number of LM experiments in the paper. We believe this has significantly strengthened the paper and are pleased that the three review responses we’ve received raised their scores by 2, 2, and 3 in light of our revisions.

Below, we summarize the key improvements from the review period:

- **Experiments with larger and smaller LMs** (in response to points raised by reviewers MMNX, 8wBr, and 4hyJ): In addition to our experiments with an 8-billion parameter LM (Llama 3.1), we added experiments with a larger 70-billion-parameter LM (Llama 3) and a smaller 1-billion-parameter LM (Llama 3.2). We found that our method enables smaller LMs to outperform models over eight times larger and allows an open-source model (Llama 3 70B) to outperform a larger, closed-source, fine-tuned model (GPT-3 Codex 002).
- **Particle experiments** (in response to points raised by all reviewers): We conducted an extensive analysis of how performance scales with the number of particles. The results show that our SMC method increases in performance faster with more particles than basic importance sampling.
- **Literature contextualization and comparison**:(in response to points raised by reviewers gr6N, 8wBr, and 4hyJ): To better link our ablation studies with existing literature, we added a new baseline method, rewrote the model descriptions, and introduced clearer shorthand notations. We also clarified our relationship with two earlier papers using SMC in related ways: Lew et al., (2023) and Zhao et al., (2024).
- **Evaluation of computational cost** (in response to points raised by reviewer 4hyJ): We included a discussion and small experiment analyzing the computational cost of our approach.

---

### Meta-Review · Area_Chair_G546 · 2024-12-23

**Metareview:**

This paper presents a novel inference-time method for controlled generation with large language models (LLMs), leveraging Sequential Monte Carlo (SMC) techniques. By extending the probabilistic framework introduced in Lew et al. (2023) and incorporating semantic potentials and resampling, the authors show how SMC-based methods can more faithfully approximate distributions over syntactically or semantically constrained text sequences. They apply their approach to four challenging tasks—Python code generation (DS-1000), text-to-SQL, goal inference, and molecule synthesis—and demonstrate consistent improvements over a variety of baselines and ablations. These gains hold across different model sizes, including smaller (1B) and larger (70B) LLMs, and across tasks that demand global semantic constraints beyond local token-level restrictions.

**Additional Comments On Reviewer Discussion:**

Strengths:
	1.	Principled Framework: The authors ground their approach in a rigorous probabilistic perspective, offering a coherent interpretation of constrained generation as sampling from a product-of-experts distribution p(x)*ϕ(x).
	2.	SMC as a Flexible Toolbox: By using SMC, they can incorporate constraints at inference time without retraining the model. This opens up a wide range of downstream applications where constraints may not have been anticipated at training time.
	3.	Strong Empirical Results: The paper shows that adding semantic potentials and resampling steps yields substantial accuracy improvements over locally-constrained decoding and simple sample-and-rerank baselines. Notably, the method scales favorably with the number of particles and can enable smaller models to surpass performance levels of much larger models.
	4.	Thorough Evaluation and Analysis: The authors have performed extensive additional experiments in response to reviewer feedback, demonstrating the robustness of their method across multiple model sizes (1B, 8B, 70B) and examining scaling behavior with the number of particles. They also offer insightful analyses linking approximation quality (KL divergence) to downstream performance.

Improvements Since Rebuttal:
	•	Wider Range of LMs: The authors evaluated their approach on smaller and larger Llama variants, and even showed that with their method a 70B open-source model can outperform a larger, closed-source fine-tuned Codex model.
	•	Particle Scaling Experiments: They conducted detailed experiments analyzing how performance scales with the number of particles, confirming that SMC benefits more from additional samples compared to importance sampling baselines.
	•	Comparisons and Contextualization: The paper now better relates to existing literature, providing a new baseline (grammar-only SMC), and clarifying connections to Lew et al. (2023) and Zhao et al. (2024).
	•	Computational Cost Analysis: The authors addressed concerns about efficiency and included a discussion on computational overheads.

Minor Comments & Suggestions:
	•	The exposition has improved, but there are still parts of Section 2 that might be streamlined further in a camera-ready version for better readability.
	•	While the authors mention that extending the method with different resampling schemes is possible, a more detailed discussion of how resampling strategies affect performance could benefit future readers.

---

### Decision · Program_Chairs · 2025-01-22

Accept (Oral)